# Recent Advances in Mesoporous Silica Nanoparticle-Mediated Drug Delivery for Breast Cancer Treatment

**DOI:** 10.3390/pharmaceutics15010227

**Published:** 2023-01-09

**Authors:** Ruma Rani, Parth Malik, Sunena Dhania, Tapan Kumar Mukherjee

**Affiliations:** 1ICAR-National Research Centre on Equines, Hisar 125001, Haryana, India; 2School of Chemical Sciences, Central University of Gujarat, Gandhinagar 382030, Gujarat, India; 3Department of Bio & Nano Technology, Guru Jambheshwar University of Science and Technology, Hisar 125001, Haryana, India; 4Institute of Biotechnology (AIB), Amity University, Noida 201313, Uttar Pradesh, India

**Keywords:** breast cancer, chemotherapeutic drugs, metastasis, mesoporous silica nanoparticles, drug delivery, pharmacokinetics, sustained release, biodistribution

## Abstract

Breast cancer (BC) currently occupies the second rank in cancer-related global female deaths. Although consistent awareness and improved diagnosis have reduced mortality in recent years, late diagnosis and resistant response still limit the therapeutic efficacy of chemotherapeutic drugs (CDs), leading to relapse with consequent invasion and metastasis. Treatment with CDs is indeed well-versed but it is badly curtailed with accompanying side effects and inadequacies of site-specific drug delivery. As a result, drug carriers ensuring stealth delivery and sustained drug release with improved pharmacokinetics and biodistribution are urgently needed. Core–shell mesoporous silica nanoparticles (MSNPs) have recently been a cornerstone in this context, attributed to their high surface area, low density, robust functionalization, high drug loading capacity, size–shape-controlled functioning, and homogeneous shell architecture, enabling stealth drug delivery. Recent interest in using MSNPs as drug delivery vehicles has been due to their functionalization and size–shape-driven versatilities. With such insights, this article focuses on the preparation methods and drug delivery mechanisms of MSNPs, before discussing their emerging utility in BC treatment. The information compiled herein could consolidate the database for using inorganic nanoparticles (NPs) as BC drug delivery vehicles in terms of design, application and resolving post-therapy complications.

## 1. Introduction

Incessantly emerging as a dreadful health concern, breast cancer (BC) remains the foremost, with 3.2 million predicted global sufferers by 2050 [1]. As of 2018, BC was attributed to 2.08 million sufferers amongst the 18.08 million cancer cases (~11.6%), culminating in more than six hundred thousand deaths out of 9.55 million cancer deaths (~6.6%) [2]. Most authentic BC screening is carried out via immunohistochemistry (IHC), recognized as estrogen receptor (ER), progesterone receptor (PR), and human epidermal growth factor receptor (HER2) positivity [3,4]. The tumors devoid of these receptors are designated as triple-negative, exhibiting typically higher invasiveness, metastasis and recurrence [5]. Currently, chemotherapy is the most effective BC remedy, although its side effects pose a serious risk to a patient’s health. Chemoresistance is an inevitable outcome of chemotherapy, resulting in impaired drug response. Figure 1 depicts the major BC chemoresistance mechanisms, exerted either via the early metabolism of administered drugs or through a lack of site-specific action (below par tumor cell penetration) [6,7,8]. Thereby, it is highly urgent to develop and optimize novel therapies comprising vehicles ensuring stealth delivery and the sustained release of delivered drugs.

Nanoparticle (NP)-mediated drug delivery presents a ray of hope in this direction, enabling improved drug stability and biocompatibility, enhanced permeability and retention (EPR), and site-specific actions [9,10]. The motivations behind using NPs as drug carriers are their robust preparation methods, self-assembly driven in vivo stability, and high surface area (SA)-driven functionalization [11,12]. Recent advances in NPs’ characterization have improved an understanding of their shape and size-modulated functional response, owing to which drug loading and release mechanisms are being better controlled [13,14]. Figure 2a demonstrates the various NPs being used for drug delivery alongwith their unique structure–function correlations concerning drug binding and transport. With swift advancements in surface characterization techniques and reliability assessments of predictive outcomes, hybrid NPs manifesting combined attributes of diverse NPs have emerged. As a result, targeting multiple tumor hallmarks minimizes the chances of non-specific actions, thereby reducing the side effects and toxicity risks to normal and healthy cells through a prolonged drug release (Figure 2b). Such attributes of NPs with a tailored tumor cell receptor-modulated surface create interest in exploring their countering of overexpressed efflux proteins, defunct apoptotic pathways, and the redox status of tumor vicinity. Keeping the above aspects in mind, this article focuses on the recent advances in the use of mesoporous silica nanoparticles (MSNPs) as drug carriers for BC treatment. Emphasis has been laid on understanding the structure–function attributes of MSNPs for explicit drug loading and delivery strategies. The MSNPs are the inorganic nanoplatforms with >700 m^2^·g^−1^ SA, >0.6 cm^3^ pore volume (PV), (2–10) nm tunable internal mesopores, and ease of chemical functionalization with a controllable degradability in biological environments alongwith high compatibility [15,16,17].

What distinguishes MSNPs from Au and Ag NPs is their core–shell morphology, wherein cores adsorb the drugs, while shells comprise aggregation guarding materials. The slightly complex and heterogeneous makeup of MSNPs, though, renders the in vivo control stiff, accounting for a likelihood of greater toxicity, albeit not alarmingly high [18,19]. Interestingly, the simultaneous use of Au and Ag NPs has been attempted, using Au and Ag nanostructures as the gate stimulus in the porous periphery of MSNPs, resulting in higher specificity and much better in vivo specificity [19,20]. High drug loading, tunable pore sizes and pore volumes, adaptable surface topologies via different SAs, and robust functionalization diversity are distinctive characteristics of MSNPs as drug carriers. The silica in MSNPs confers them manifold functional diversification attributes, such as thermal stability, biocompatibility, easier modulated chemical aspects, and facilitating controlled drug release at the targeted site. The functionalization efficacy engineers could reduce the chances of accidental misfired targeted delivery by simultaneous interactions with more than one tumor cell surface receptors. Most of these characteristics do not prevail optimally in Au and Ag NPs, due to which we intend to discuss in their prominent contributions in the drug transport efficacy of MSNPs. These features encompass the site-specific drug delivery, besides the in vivo structural protection of drugs, accomplishing higher tumor cell internalization at much lower intakes. The selection of MSNPs as carriers has been made in order to understand their therapeutic distinctions with the relatively much hyped Au and Ag NPs, conceptualized via our previous critics [21,22,23]. The information comprised herein could advance the future design of accurate and safe drug carriers for solid tumors. 

## 2. Exclusive Terminologies and Synthesis Methods of Mesoporous Silica Nanoparticles

Structurally and morphologically, the MSNPs are solid entities with pores enriched on their surface. “Mesoporous” in MSNPs was a term used to describe pore size. As per the International Union of Pure and Applied Chemistry (IUPAC) regulatory guidelines, porous materials may be microporous, mesoporous, or macroporous. Although their synthesis dates back to the 1970s, Mobil Research and Development Corporation created the first mesoporous solids from aluminosilicate gels in 1992. The associated industrialists labeled it as “Mobil Composition of Matter” or “Mobil Crystalline Materials” (MCM) [24,25]. Relative distinctions amongst micro, meso, and macroporous are primed exclusively vis-à-vis pore diameters, with <2 nm, (2–50) nm and >50 nm domains. The prominent drug carrier-suited aspects of MSNPs comprise their alterable particle and pore sizes, high stability, rigid structure, large SA, high PV, and robust functionalization and fabrication methods [26,27].

MSNPs belong to the Exxon Mobil-based M41S group functional configurations, prepared under alkaline conditions in the presence of alkylammonium surfactants and a 3–10 nm pore-sized silica source [28]. The functional physicochemical aspects of MSNPs have been demonstrated as significantly varied, via specific choice of initiating precursors, template agents, and reactions. Characterizing aspects of these are their pore sizes (permeability regulators), overall morphology, pore volume and specific SA. Studies demonstrate MCM-41, MCM-48 and MCM-50 as the prominent variants of the M41S family, characterized, respectively, by a hexagonal arrangement, cubic configuration and lamellar morphology [26,29]. MCM-41 is fabricated using cationic surfactants within a 8.5–12 pH range [30]. The most common pore architecture is hexagonal, wherein the cationic-surfactant-to-silica-source (CSS) proportion remains <1 [31]. Similarly, MCM-48 and MCM-50 were made by varying the CSS stoichiometry. A cubic pore structure of MCM-48 was retrieved when the CSS ratio was >1 [32]. When the stoichiometry was increased further, MCM-50 gradually provided pore structures that resembled lamellar sheets [29]. Other mesostructured materials such as Santa Barbara Amorphous (SBA) types were synthesized at the University of California and Technische Universiteit Delft (TUD-1) was developed at Delft University of Technology; COK-12 was synthesized at the Center for Research Chemistry and Catalysis; HMM-33 (Hiroshima Mesoporous Material-33, Japan); and KIT-5 was synthesized at the Korean Advanced Institute of Science and Technology having varied pore sizes and symmetry [33,34,35]. Compared to other members of the MCM family, SBA exhibits a hexagonal 2D structure with thicker pore walls and larger pore size. The SBA mesostructured materials, such as SBA-11, SBA-12, SBA-15 and SBA-16, were created using non-ionic triblock copolymers as templates, such as poly(alkylene oxide) block copolymers and alkyl poly(ethylene oxide) (PEO) oligomeric surfactants [36]. These materials prevail in cubic, 3D hexagonal, 2D hexagonal and cubic cage pore symmetry.

### 2.1. Synthesis Methods

Preparation of MSNPs is characterized by the requirements of (i) the silica precursor, the SiO_2_ source, which forms the walls around the pores, (ii) a surfactant, which aids in the formation of pores, and (iii) an acid or base catalyst, which controls the morphology of the silica structures. The inception of MSNPs’ formation emerges from the sustained interactions of surfactant and silica sources, which provides mesoporous textures. The first successful MSNP synthesis was reported by the Cai, Mann and Ostafin groups, wherein the MSN acronym was coined by Victor Lin to signify the mesoporous silica nanospheres [37,38,39,40]. Common SiO_2_ sources used herein include tetraethyl orthosilicate (TEOS), tetramethyl orthosilicate (TMOS), tetra methoxy vinylsilane (TMVS), tetra butoxy silane (TBOS), tetrakis (2-hydroxyethyl) orthosilicate (THEOS), tetra propyl ortho-silicate (TPOS), trimethoxy silane (TMS) and sodium metasilicate (Na_2_SiO_3_). Noted catalysts in the process include NaOH, HCl, NH_3_, diethanolamine (DEA) and triethanolamine (TEA) [27,41,42,43,44,45,46,47,48]. The surfactant used in the MSNPs’ synthesis is the reason for the varied morphologies, from dispersed monospheres to agglomerates, manifested from altered TEOS hydrolysis and CTAB micellization.

The prevalence of agglomerates in the solution phase relates to a threshold surfactant extent, which varies with the mutual hydrophobicity of the silica precursor. Familiar surfactants used as templates in MSNPs are cetyltrimethylammonium chloride (CTAC), cetyl trimethyl ammonium bromide (CTAB), hexadecyltrimethylammonium (HDTMA), sodium dodecyl benzene sulfonate (SDBS), sodium dodecyl sulfate (SDS), Triton X-100, polysorbate, Pluronic F127, Pluronic P123, and phospholipids, betaines and amino acids [49,50,51,52]. The key parameters affecting the formation of MSNPs are the reaction mixture’s pH, the nature of the surfactants/copolymers, in addition to the relative extents and sources of silica. Prominent synthesis methods are hereby distinguished as sol–gel and hydrothermal and a contemporary green method, the working aspects of which are discussed ahead (Figure 3). 

#### 2.1.1. Sol–Gel Method

The majority of MSNPs are synthesized using a modified Stober’s approach, commonly recognized as the sol–gel approach, having been extensively studied for the synthesis of numerous inorganic compounds. To create an ordered network of polymer or discrete particles, the hydrolysis and condensation of alkoxide monomers in a colloidal solution are needed [53]. During hydrolysis, reactive silanolates (=Si-O-) are formed, which condense with more of their population to form covalent siloxane bonds (Si-O-Si) with increasingly larger oligomers (Equations (1)–(3)). Typically, an acid or a basic catalyst sustains a sol–gel process. When hydrolysis proceeds faster than condensation, an acidcatalyst is used, generating a large number of tiny SiO_2_ particles or a web of gels. On the contrary, when condensation is rapid, a base-catalyzed reaction provides particles of solid sphere morphology [36].

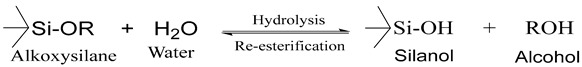
(1)

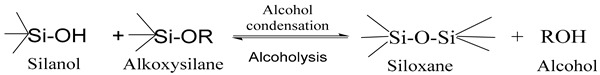
(2)

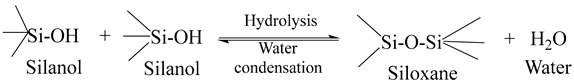
(3)

The first reaction illustrates the formation of silanol groups (Si-OH) via alkoxysilane hydrolysis, wherein an –OH group replaces the alkoxide (-OR) group (Equation (1)). The rate of this reaction is affected by the water–alkoxide stoichiometry, working pH, and the nature of the solvent and catalyst. The second and third reactions describe the condensation of as-formed silanol groups, with an alkoxide or more silanol groups, giving siloxane bonds (Si-O-Si) and alcohol/water as byproducts (Equations (2) and (3)). Our literature survey for MSNP synthesis revealed the extensive use of the sol–gelmethod [54,55,56,57,58]. 

#### 2.1.2. Hydrothermal Method

The terminology “hydrothermal” was first used by Sir Roderick Murchison, a British geologist, to refer the formation of minerals amidst the rise of hot water solutions from cooling magma. Chemical scientists define hydrothermal synthesis as a reaction that happens in a sealed container at a high temperature and pressure. Typical modifications in the process comprise the reorganization of mesostructures, followed by their growth and crystallization, in the course of the solution reaction. The adsorptive and structural traits of SBA-15 and SBA-16 can be varied by changing the duration and temperature of a typical hydrothermal treatment. The SBA-15 materials with larger mesopores and thinner pore walls require high temperatures or longer hydrothermal treatment than samples prepared under standard conditions [59,60]. In a significant attempt, MSNPs were prepared using CTAB and TEOS in the presence of NH_4_OH over a 12 h chemical conditioning.The (CTAB + TEOS) solution was poured into a high-temperature kettle, allowing the reaction to occur for 3 h in an oven at 180 °C [61]. Even though the method formed particles of a mesoporous texture, it requires specialized equipment and a high temperature for a long time, accounting for the method’s high cost, making sol–gel approach the preferred regime.

#### 2.1.3. Green Synthesis

This method has recently emerged as one of the paradigm shifts for MSNP synthesis via the green route. The method uses every agricultural waste that has a significant content of silica as a precursor to synthesize silica components and uses plant metabolites as a surfactant [62]. Different kinds of agricultural waste, for instance, horsetail plant [63], hexafluorosilicic acid (a fertile waste) [64], banana peel ash [65], rice husk [66], palm kernel shells [67] and bagasse ash [68], have been used as a silica source for the preparation of MSNPs. In another study, β-glucosidase was immobilized on tannic acid-templated MSNPs to synthesize biocatalysts. The results showed the catalyst exhibited good thermal, catalytic and operational stability with a large quantity of tannic acid [69]. A first look infers no green chemistry in the adopted route, although the method is benign. For example, Na_2_SiO_3_ extraction from wastes such as ash, rice husk, and others, is cumbersome, but its formation increases at extremely low and high pH.

## 3. Promising Aspects of Mesoporous Silica Nanoparticles as Drug Vehicles

The drug delivery attributes of MSNPs are majorly inspired by their functionalization-driven interactive potential. The flexible surface morphology is the reason for shape and size-manifested physicochemical activities, paving the way for tunable biophysical performance. The robust mechanisms for tuning the MSNPs’ interactive activities as drug carriers comprise the shape, PS and pore size. Diverse forms of silica (chemically, SiO_2_) primed for drug delivery comprise fumed porous and non-porous textures. Multiple investigations at a preclinical scale employing silica-based drug carriers have demonstrated significant improvements in moderating the dosages and improving the extent of tumor cell internalization. However, commercial readiness to use these materials as drug vehicles is still under investigation, wherein the determination of in vivo toxicity emerges as decisive. Pertaining to shape–size-modulated physicochemical changes (biological sensing) and drug delivery efficacy, silica NPs are generally amorphous and could be porous or non-porous. The preferential suitability of an amorphous texture is attributed to the higher entropy, contrary to the crystalline forms, clearance of which is a big hurdle, accounting for uncertain in vivo biochemical activities [70]. Studies postulate a swift dissolution of MSNPs at below saturation extents, wherein a study by Martin and collaborators demonstrated the bio-fluid dissolution of silica, subsequently obtaining it absorbed or excreted as silicic acid (via urination) [71]. Silica NPs for drug delivery rapidly undergo degradation to silicic acid via hydration, hydrolysis and ion exchange. This degradation relieves the toxicity that could be due to the deposition of silica NPs and is precisely affected by the nature of the medium and the concentration of the relative NPs. To expedite this degradation, the non-covalent doping of organic moieties and the covalent binding or incorporation of cleavable organically bridged silsesquioxanes into silica NPs are noted mechanisms. It must be noted here that the explicit degradation of MSNPs is more complex than silica NPs, due to the altered matrix chemical composition [72]. Such distinctions are the outcomes of the differential rate and condensation extents of silica matrices amidst the sol–gel-driven synthesis. The following paragraphs elucidate the impact of salient physicochemical properties (PCPs) affecting the MSNPs drug delivery potential. These parameters prominently include the particle shapes (morphology), size and surface chemistry. Moderate binding with carriers, alongwith the relative extent of surface engineering, tunable pore sizes and PVs, are some prospects which make MSNPs distinct drug carriers 

### 3.1. Surface Chemistry

In light of the conducted investigations, silanol groups on the silica surface are the decisive contributors to silica toxicity, which promptly interact with membrane components that culminate in the cell lysis and, ultimately, the disorganization of cellular constituents [73,74]. The porous morphology in MSNPs is exclusively responsible for a comparatively lower hemolytic effect compared to that of non-porous morphology. In this regard, structure–function analysis deciphers the role of the low density of surface silanol functionalities, which critically affect their biodistribution and biocompatibility. For instance, modifying the surface receptivity via polyethylene glycol (PEG) functionalization augments the stealth delivery of MSNPs, minimizing the deposition in the liver, spleen and lungs. The source of this in vivo risk minimization is PEG biocompatibility, which prolongs the circulation time of engineered MSNPs [75]. In a noted effort, Yu and colleagues examined the impact of surface chemistry on MSNPs’ toxicity in macrophages (RAW 264.7) and A549 lung cancer epithelial cells. They found that amino-modified mesoporous SiO_2_ showed a significantly higher cellular association than their bare mesoporous counterparts [76]. 

In a previous attempt, Townson and colleagues studied the effect of spatially arranged surface –NH_2_ groups in MSNPs of a similar size, surface charge and porosity. Modification with trimethoxysilylpropyl-engineered polyethyleneimine (PEI), 2-[methoxy(polyethyleneoxy)propyl] trimethoxysilane and N-trimethoxysilylpropyl-N,N,N-trimethyl ammonium chloride provided PEG-PEI and PEG-NMe_3_^+^ MSNPs with an exposed and hindered –NH_2_ functionality. Monitoring the cytotoxicity in vitro and in vivo, the –NH_2_-functionalized NPs’ activities were analyzed on A549 (human lung cancer), A431 (human epithelial cancer), Hep3B (human hepatocellular cancer) and human hepatocyte cells. The analysis revealed PEG-PEI-functionalized NPs were bound to all cells, while those functionalized with PEG-NMe_3_^+^ exhibited a restrictive activity [77].

### 3.2. Effect of Varying Sizes

Regarding size limit, numerous misconceptions prevail in the existing literature and the imperative understanding of these accounts for the fact that smaller dimensions lead to dominant surface effects (greater quantum confinement). The usefulness in accurate diagnosis and site-specific activities indeed requires a prompt response and high surface payload, but small particle size (PS) is not the sole criterion for these requirements. This is because smaller dimensions imply a weaker control of gravity, owing to which, engaging in non-specific biochemical reactions is quite high. In general, though, particles < 10 nm in size are easily eliminated by the renal route, while those >400 nm penetrate tumor cells with great difficulty. Thereby, retaining caution regarding the unpredictable chemical and physical activities conferred by enhanced SA, the (50–100) nm range is considered the optimum for drug distribution and tumor cell uptake [78]. Manifold strategies are known to arrest the MSNPs’ PS, with one investigation linking the hydroalcoholic and aqueous medium stoichiometric balance alongside a higher hydroalcoholic proportion to a larger PS. The same research group also associated a >7 pH (higher alkalinity) with a low PS distribution [79]. 

Another study demonstrated the possible variability in the PS by changing the triethanolamine (TEA) amount, with a reduced quantity giving a smaller size [80,81]. Surprisingly, this generalization was contradicted by the 2019 and 2016 attempts of theIsa and Lv groups [62,82]. In a further attempt towards attaining a stable PS, Yismaw and colleagues screened the effect of various catalysts on the retrieved PS via the analysis of NPs prepared using NH_3_ and TEA as catalysts. The inspection revealed that NH_3_ as a catalyst provides highly agglomerated spherical particles with an average size of 420 nm, but on changing the catalyst to TEA, a reduced PS of 186 nm was obtained. Such distinctions led to the conclusion that using TEA as a catalyst supports the bottom-up mechanism, enabling a growth arrest vis-à-vis the optimum stoichiometry [83]. Apart from TEA and NH_3_, several subsequent research attempts list NaOH as a suitable catalyst, illustrating its growth control ability, with a higher pH being instrumental in obtaining a low PS [84,85,86]. Another potential way to control the MSNPs’ PS is via template variation. Earlier studies have used tannic acid, gallic acid, eudesmic acid, ethyl gallate and quercetin as templates, exhibiting a structural similarity in being active H^+^ donators (Figure 4). One of these attempts herein noticed the exclusive formation of pores using only tannic and gallic acids as templates. This study observed that replacing tannic acid with gallic acid as a template increases the PS from 240 to 600 nm [87]. This distinction could be due to manifold –OH groups and tannic acid cavitation, which are quite meager in gallic acid (four–OH groups and no cavitation possibility). 

Steric stabilization by longer alkyl chains of the capping agent could also be a strategy to obtain a low PS [88,89]. Other significant efforts in this direction noticed a formation of larger-sized particles at higher temperatures, although the intermolecular forces are weakened with increasing temperatures due to the enhanced kinetic energy of the molecules [90,91].

### 3.3. Morphology or Particle Shape

Concerning NPs’ interactive potential (cellular uptake and biodistribution patterns), the characteristic particle shape remains a decisive factor affecting the therapeutic efficacy. For instance, the rod regime finds it comparatively cumbersome to invade a tumor cell rather than the spherical one, which remains the most studied to date. The working mechanism for distinct morphologies relies on varied TEOS and 3-aminopropyltriethoxysilane (APTES) stoichiometries, with a higher value for a rod shape [92]. A pertinent effort in this direction reported yolk–shell-resembling MSNPs, embedded with organic phenolic resin as a template, which was removed via calcination to form porous silica NPs [93]. A relatively uncommon morphology resembles a typical viral core, featuring a singular micelle-driven epitaxial growth in a dilute surfactant oil–water biphasic system. Utilized MSNPs had mesoporous silica in the core with peripheral silica nanotubes at a right angle to the surface [94]. Another recent study by Isa and colleagues used pyridinium ionic liquids with anions as a template, providing spherical, distorted spherical and raspberry morphologies [95].

### 3.4. Pore Diameter

The typical pore size of MSNPs ranges from 2 to 3 nm, which is the optimum for the in vivo trafficking of small molecules [96]. Disappointingly, the porous nature does not synchronize well for large molecules, which further diminishes their use as drug carriers. Consequently, research attempts have been made to expand the pore size of MSNPs, wherein one study reported the adjustment of varying the TEOS concentration in cyclohexane. The optimization provided MSNPs with pore sizes of 7.8–12.9 nm [97]. In an earlier effort, Liu and group prepared MSNPs resembling the dendrimer morphology, with the nanodimensional, central radial pores and exhibiting an increment in pore size from the core to the NP surface. The as-prepared MSNPs were the outcome of a biphasic stratification reaction system, comprising a chlorobenzene–water mixture with a 22 nm pore size [98]. Another significant effort reported larger pore sizes by varying the ethanol-to-water volumetric proportions. The analysis revealed a decrement in this ratio for an enhanced pore size of 6.52 nm that relied on the competitive interactive behavior of ethanol and water [99]. In a more robust regime, Ryu and associates designed MSNPs with large pores via the rapid cooling of the reactant, followed by reaction continuation. The fabricated MSNPs exhibited a wrinkled morphology with pore sizes in the range of 15–33 nm [100]. To increase the pore size, several studies have used a pore expander. In distinct efforts, a pore size of 5.3 nm was obtained using trimethyl benzene as a pore expander; while using tri-isopropylbenzene, >10 nm pore sizes could be obtained [101,102,103]. The functioning of a pore expander is based on a proportional relationship between pore size and the quantity of pore expander added [104]. Thus, varying the solvent-to-water ratio in the reaction medium, optimizing the new synthesis environment and use of a pore expander could be potential mechanisms to obtain a large pore size. 

### 3.5. The Gatekeeping Concept

Gatekeeping modulation comprises the subjection of a cell to a capping agent that responds to internal, external, physical, or chemical stimuli, driving the complete or partial opening of MSNPs pores, enabling a sustained drug release without aggregation [105]. Several gatekeeping agents, including proteins [106], amino acids [107], polymers [108], peptides nucleic acid [109] and macrocyclic molecules [110], have been used to forbid the premature drug release from the NPs [111]. The controlled release of loaded drugs from MSNPs is due to the gatekeeping functionalization using diverse stimuli such as light, temperature, pH, redox activation and enzymes [112,113]. For example, a low-molecular-weight gelator with dual pH and glucose-sensitive moieties was synthesized by Gao and team, and then phenethylbiguanide-loaded MSNPs were penetrated by the gelator. In the resulting formulation, the release of phenethylbiguanide could be controlled via changing the gel sensitivity to the stimuli and was responsive to glucose levels and pH [114]. In a subsequent attempt, Zhang and associates synthesized bioresponsive controlled delivery systems using the structure-switching adenosine aptamer. The assembly and disassembly of Au NPs from MSNPs were tuned by the aptamer, leading in an adenine -triggered controlled release [115].

### 3.6. The Flexibility of Drug Loading

This attribute of drug delivery with MSNPs relies on their adsorption ability, which in turn is due to their large SAs and is further aided by the functionalization for the loading of hydrophilic, as well as hydrophobic, drug cargos. Large PVs facilitate the loading of higher drug quantities, which is a distinct feature of MSNPs. The loading of drugs could be accomplished either in situ (amidst their preparation) or via fresh encapsulation onto the synthesized MSNPs. Multiple studies illustrate significant modifications in the synthesis methodology whereby efficient loading could be accomplished. For instance, a study by She and colleagues demonstrated the 5-fluorouracil (5-FU) loading over the hollow MSNPs via functionalizing the silanol groups with octadecyltrimethoxysilane (OTMS), (3-aminopropyl) triethoxysilane (APTES) and 3-cyanopropyltriethoxysilane (CPTES). Figure 5 depicts the chemical structures of OTMS, APTES and CPTES, with the highest steric stabilization for 8 carbon alkyl chain (C-AC) and common SiO_3_ functionality. 

This distinction argues for a monodispersed prevalence for OTMS functionalization, where the geometry varies drug loading, distinguished via28.89% when hollow and 18.34% in solid state. Investigators reasoned for an efficient loading vis-à-vis the hollow morphology via 5-FU and cationic –NH_2_ group-modified MSNPs, with electrostatic interactions. Enhanced loading could be attained by changing the functionalization [116]. In a separate study, Wang and colleagues reported a greater drug loading before functionalization than after grafting [117,118]. The investigators also discussed a 3–15-fold greater drug loading with hollow MSNPs post functionalization, probably due to preferential cavitation. This was ably supported by a >two-fold ibuprofen and doxorubicin (DOX) loading on the hollow MSNPs than the solid morphology of MCM-41 [119,120]. 

Another mechanism for enhancing the drug loading in MSNPs could be via polymer gatekeeping, whereby low-bioavailability drugs could be entrapped [121]. This method does not require any chemical modification of MSNPs and provides a large empty PV in their unmodified state for drug encapsulation. Additionally, this approach allows a robust functionalization of a polymer shell with a targeted ligand that optimizes the delivery specifically to the cancer cells on the subsequent addition of a thiol-conjugated ligand. One potential study noticed that consecutive drug loading enhances molecular interactions, thereby accomplishing a high loading [15]. Improving the drug–feed ratio also varied the drug loading efficacy of MSNPs [122]. The primitive factor associated with drug loading is the PV, which could be expanded to carry a larger drug cargo. In this regard, several studies have demonstrated the use of alkanes/ethanol, triisopropyl benzene (TIPB), trioctylamine (TOA), decane and N,N-dimethyl hexadecylamine (DMHA) to aid pore expansion [123,124].

### 3.7. Drug Release from MSNPs

The release of drugs from MSNPs is accomplished by diffusion across the pores and can be modulated via surface functionalization to meet specific biological needs. Inevitably, the major factor regulating drug release is the binding strength of pore surface groups (in MSNPs) and the loaded drug(s). Ideally, the binding forces (BFs) must be non-covalent, which should not induce any drastic structure-breaking effect [125]. Release extent also varies with pre- or post-functionalization drug loading, with the latter mode sustaining the release via the restrictive influence of functional groups (mostly noticed for –NH_2_ groups). Studies reporting this distinction for pre-functionalization drug binding correlate this with APTES capping inhibited drug release from the pores. In the event of a prior functionalization, the drug loading may result in its surface adsorption, culminating in its burst release. In a rigorous attempt, Wang and colleagues reported APTES-modified MSNPs which worked via charge reversal, exhibiting varied drug release with varying APTES concentrations [117]. A 2020 study studied the drug release of 5-fluorouracil (5-FU) and dexamethasone (DMX) via loading on an anionic surfactant and APTES (as template)-carrying MSNPs. The –NH_2_ functionalization of MSNPs enabled 83% (for 5-FU) and 21% (DMX) release in 48 h at 7.4 pH, the differences being attributed to the distinct hydrophilicities of 5-FU (surface adsorption) and DMX (binding within the pores via electrostatic interactions). Moreover, the –NH_2_ functionalization of NCs was instrumental in their compatibility within (0–468) μg·mL^−1^ screened via 48 h incubation with HT-144 cells [126].

In another elegant attempt, aspirin loading and release from MSNPs was monitored via synthetic grafting, and co-condensation was demonstrated, wherein the latter approach enabled a higher drug release. Working on plain MCM-41, aspirin and silanol groups, moderate interactions enabled a prompt drug release obeying Fick’s diffusion. However, the –NH_2_ functionalization in MCM-41 delayed the aspirin release to a higher extent in co-condensed MCM-41 [127]. Almost along the same lines, a Santa-Barbara-Amorphous-type-material (SBA-15)-bound ibuprofen release exhibited significant variation, with SBA-15 surface engineering. The analysis revealed a wide variation in ibuprofen release from the –NH_2_-functionalized SBA-15, with a 100% release in 10 h compared to the three days for post-synthesis drug loading [128]. Even when loaded before functionalization, the interactions with functionalizing moiety are a prominent factor regulating drug release. Figure 6 depicts thechemical structures of aspirin, silanol and ibuprofen, wherein diffused pi-conjugation and connected hydrophilicity in aspirin and ibuprofen project their additive interaction sensitivity. With manifold H and –OH groups, silanol exhibits greater hydrophilicity and seems receptive towards aspirin due to bothits substituted moieties (Figure 6a,c). Contrary to this, ibuprofen has a hydrophilic sensitivity only at one substituent, and, therefore, its binding with silanol is controlled by weaker interactions (a positive aspect for delivery) than aspirin. Simulations like this are accurate predictors of in vivo carrier performance, flanked by the optimum strength of relative BFs by which its drug release could be controlled and optimized.

A broader attempt in 2022 further adds to the structure–function correlation for MSNPs, wherein monolithic morphologies were examined for their release of anastrozole (used for BC), imiquimod, levetiracetam, xylazine and flunixin. Quite intriguingly, PVs and initial loading amounts were screened as the decisive factors regulating an adequate drug release. No significant release variations were observed with the seemingly sensitive pore diameter and surface area. This study screened the effect of PVs on the release extent via statistical correlation and diffusion coefficient variations. Most surprisingly, the analysis revealed a greater drug release for large PVs (0.17–0.27 cm^3^·g^−1^) within 60 min, whereas for lower PVs (0.13 cm^3^·g^−1^), the maximum release extent was attained in 150 min. Enhanced mesoporous sensitivity with smaller pore diameters and 0.13–1.12 cm^3^·g^−1^ PVs implied a direct correlation of drug release on PVs and predicted a significant linear correlation between the diffusion coefficient and PV [129]. This study analyzed the transdermal release efficacy of the drugs, and since the skin environment exhibits significantly distinct permeability than tissues within the body, these results might not be compatible for drug delivery to solid tumors. 

### 3.8. Improved Drug Pharmacokinetics with Mesoporous Silica Nanoparticles 

The versatile surface chemistry of MSNPs allows us to engineer their adsorption attributes, making them feasible carriers of hydrophobic drugs. To elucidate this, Bukara and colleagues attempted to solubilize the drug fenofibrate, followed by an assessment of its therapeutic efficacy in healthy human volunteers [130]. Screening of the volunteers, four days after fenofibrate administration, revealed the presence of fenofibric acid (a pharmacologically active metabolite) in their plasma. Strikingly, a 177% C_max_ increment with a reduced t_max_ was noticed in response to the orally administered fenofibrate formulation. In another significant attempt, Zhang and associates delivered telmisartan (TEL) via loading over the MSNPs and studied the pharmacological impact in beagle dogs. The in vitro efficacy of MSNP-delivered TEL was inferred by its enhanced uptake in Caco-2 prostate cancer cells, while the in vivo impact was screened by a 1.29-fold enhanced area under the curve (AUC) post 72 h of MSNP-delivered TEL compared to its marketed tablet and blank MSNPs [131].

In a separate study, Thomas and colleagues loaded their synthesized MSNPs with carbamazepine (CBZ), oxcarbazepine (OXC) and rufinamide (RFN) antiepileptic drugs. The dissolution kinetics of these drugs in phosphate buffer exhibited a faster CBZ and OXC release compared to an initial burst regime for RFN. Figure 7 depicts the chemical structures of CBZ, OXC and RFN, with a greater similarity between CBZ and OXC, while the RFN has a fluoride substitution in its terminal benzene ring. The prevalence of accessible terminal amide functionality could be a common factor associated with their usage similarity, but an initial burst release for RFN could be due to its weaker or surface-restricted adsorption over the MSNPs [132]. The above studies highlight the functionalization-modulated drug binding and release kinetics for MSNPs, whereby the differential chemical structures could be exploited alongside moderated instantaneous patient sensitization.

## 4. Drug Release Mechanisms of Mesoporous Silica Nanoparticles

As is the case with most drug-delivering NPs, MSNPs could also be tuned for active or passive drug delivery. The former of these mechanisms relies on the proximity to surface ligands, while the passive regime exploits the troubled physicochemical vicinity of tumor cells for tumor cell drug internalization. Figure 8 depicts the functional distinctions of ligands frequently used for MSNP-aided drug delivery to tumor cells. It is worth noting here that a delivery regimen could be programmed via dual ligand binding, depending on the tumor severity. Contrary to the above, the passive mode relies on enhanced permeation and retention (EPR), gathering the threshold drug extent in the tumor cell vicinity. Although the active mode is more accurate, its optimization is comparatively stringent (ligand topography and distribution), alongwith a greater risk vis-à-vis localized pH or temperature changes. Several comprehensive reviews discuss the mechanistic distinctions of active and passive delivery regimes, pictorially distinguished in Figure 9. Unlike Au and Ag NPs, MSNPs exhibit significant functionalization diversity, whereby the role of surface pores, gatekeeping-driven drug release and PVs together assume a significant contribution. 

These distinctions provide the MSNP-mediated drug delivery with the possibility of multiple stimulus-driven performance modulations. Although these controls are empirically the eventualities of the functionalization modes, the explicit terminological distinctions for MSNPs are quite significant for understanding them thoroughly with regard to their drug delivery potential. This is often discussed via the shape–size-modulated physical response, assuming a decisive role in tumor cell internalization and sustained drug release, capping agent binding and ligand targeting. The following paragraphs are, thereby, devoted to the prominent and exclusive traits of MSNPs for their drug delivery ability.

### 4.1. Stimulus Response

Delivering drugs using MSNPs is optimally distinguished via modulation using stimulus-responsive conditioning. According to our literature search, the typical configuration of such a carrier system comprises active and passive targeting with a gate-triggered drug cargo entrapment. Both active and passive targeting exhibit trivial uncertainty, characterized by a premature cargo release during circulation, easily resolved via gate-stimulus-driven drug penetration in the MSNPs. On reaching the target sites, this gate opens in response to variations in pH, temperature, enzymes and redox status, among other things (Figure 10). Based on the number of stimuli, these mechanisms are distinguished as single, dual or multiple. The administered stimulus could be in situ or externally optimized, as summarized in Figure 11. Noted external stimuli are thermal energy via temperature enhancement, irradiation via visible and higher energy waves or ultrasound-assisted drug carrier detachment. 

#### 4.1.1. Internal Stimulus

Several internal cues are well-known and have been demonstrated to load the drug molecules to ensure a site-specific release for cancer therapy. The familiar response factors for these mechanisms include pH, redox status and enzymatic actions. 

(i)pH stimuli-responsive system

A pH stimuli-responsive drug delivery system (DDS) is optimally equipped for biomedical applications. For tumor-targeting therapy and similar illnesses where the pH of the affected area is slightly more acidic than normal tissues, pH-responsive MSNPs have demonstrated promising outcomes. Gatekeeping molecules, such as acid-degradable polymers, polyelectrolytes, pH-sensitive linkers and others, are protonated by small pH changes, which alter the molecule’s overall solubility, enabling thedrug release at a particular site. In a noted attempt to illustrate this, Lee and colleagues investigated the DOX delivery using poly-L-lysine-gated MSNPs, noticing a pH decrement and polymer-swelling-mediated drug release [133]. Another significant attempt from Chen and team created pH-responsive MSNPs with “release–stop–release” tenability using ZnO quantum dot dissolution in acidic conditions to assist the drug release. The investigators claimed that the system could minimize the traumatic side effects of residual drug contents, due to its efficient discard [134,135]. In another notable effort, He and colleagues used pH-responsive, polyacrylic acid (PAA)-modified MSNPs as andographolide (AG) carriers, forming AG@MSNPs-PAA, a nanoscale pH-responsive vehicle for osteoarthritis treatment [136]. Yet another effort by Wagner and colleagues demonstrated the pH-responsive functioning of MSNPs for the delivery of the immune modulator R848 to antigen-presenting cells (APCs), exhibiting significant anticancer potential [137]. In aprevious effort, Zhang and team noticed a synergism of arsenic trioxide (ATO) and paclitaxel (PTX) via MSNP-mediated delivery to the MCF-7 BC cells, causing a higher cell cycle arrest and apoptotic activation than the singular treatment. The optimized MSNPs exhibited PAA coating and a pH-modulated, lipid-sensitive F56 peptide targeting for a pH-modulated drug release. Interestingly, PTX release from the NC was driven by the instability of cholesteryl hemisuccinate (a vital PSL constituent) at the tumor microenvironment’s pH. A striking coincidence was the PAA protonation at <5.5 pH, which likely assists a pH-modulated drug release. Studying the extent of release at pH 7.4, 6.5 and 5.5, the investigators noted that PAA layering and lipid coating moderated ATO liberation with a 55.6% PTX and 3.8% ATO release at pH 7.4, upon 48 h monitoring. The pH 6.5 catalyzed a faster release, with 77.1 and 63.1% for PTX and ATO, respectively, possibly due to lipid disruption. While PAA layering inhibited ATO release at pH > 5, at pH < 5, 76% ATO and 97% PTX were liberated, due to the combined effects of lipid disruption, PAA and –NH2-driven MSNP- dissociation and hydrated polymer contraction on the partial protonation of –COO- functionality [138].

(ii)Redox stimuli-responsive system

Another mechanism of drug delivery for MSNPs exploits the generation of reactive oxygen species (ROS) at higher contents in malignant cells due to manifested genetic mutations, mitochondrial malfunctioning and an altered metabolism. To combat this, tripeptide glutathione (GSH) was subsequently screened as the most efficacious ROS scavenger. Though GSH restores the native redox status in normal cells, in malignant cells, it fuels the tumor development by aggravating radio- and chemoresistance. A frequently used linker to optimize the redox-responsive drug delivery using MSNPs is the thiol conjugation that is cleaved by GSH. In a recent study, Au NPs were demonstrated to serve as gate controls in the MSNPs, which were subsequently evaluated for their pH-modulated drug trafficking to BC cells [139]. As the carrier becomes exposed to an environment with a high GSH concentration, the thiol linker conjugating the Au NPs to the MSNPs’ surface is disintegrated, triggering the DOX release. Another rigorous attempt by Li and team demonstrated the gatekeeping function of coated polymer (host) layers to deliver the drug cargo in the presence of GSH via host–guest interactions. The scientists observed a varied release profile depending on the functionality of polymer-coated layers [140]. Relying on similar host–guest interactions, Xu and colleagues described a reduction-driven drug carrier composed of a supramolecular PEGylated pillar[5]arene (host) and viologen stalk (guest) nanovalve to encapsulate drugs within the MSNPs’ pores. The working mode comprised guest viologen stalks encircled by PEGylated pillar[5]arene, which became separated on encountering the reducing agent [141].

(iii)Enzyme stimuli-responsive system

The enzyme-responsive MSNP carrier utilizes enzymes as a stimulus and exhibits feasibility under mild working conditions, alongwith a high chemical selectivity and substrate specificity. This drug delivery mechanism by MSNPs uses a specific enzyme overexpressed in the tumor microenvironment and a linker that exclusively responds to this enzyme, enabling it to release the bound drug at a targeted site. In a noted effort, Jiang and teammates reported hyaluronic acid (HA)-conjugated MSNPs for 5-FU delivery vehicles to HT29 colon cancer cells. The HA conjugation on an NC aided its drug release, owing to its sensitivity towards hyaluronidase in the in vitro environment. The HA conjugation of 5-FU-loaded NC subjected 43% of cells to early apoptosis, while 55% of cells prevailed in the late apoptotic phase, exhibiting a three-fold greater potency than free 5-FU and a two-fold greater efficacy than the non-HA-conjugated state [142]. Another worthwhile attempt by Zhou and colleagues demonstrated enzyme-responsive MSNPs with HA and collagen I (extracellular matrix components) as the shell constituents. The exclusive benefits of these ECM macromolecules comprised their controlled and specific biodegradation by hyaluronidase (HAase) and matrix metalloproteinases 2 (MMP-2), prominently overexpressed in some tumors. The distinctive control exercised by varied HAse and MMP-2 expression on cancer cells enabled a 75% DOX release in 72 h, compared to the insignificant amounts without their functionalization [143]. In a more recent study, Cai and colleagues demonstrated the enzyme-responsive MSNPs with chitosan (CTS) trapped drugs inside the pores. The inspection revealed broken MSNP–CTS–azo linkages in the presence of colon enzymes [144].

#### 4.1.2. External Stimulus

To modify the drug delivery from MSNs, several additional stimuli, including light, temperature, ultrasound, magnetic and electrochemical energy-driven energization, have also been studied (Figure 10). The following paragraphs discuss the salient aspects of these mechanisms.

(i)Light stimuli-responsive system

To design light-responsive MSNPs, functionalization using one or more photoactive groups is attempted and optimized via varied wavelengths such as ultraviolet (UV), visible (Vis) and near-infrared (NIR). In a recent study, Wang and associates observed a UV light exposure-driven altered azobenzene conformation, aiding the cyclodextrin-bound DOX release [145]. Another significant study used ruthenium complex as a gate to seal the MSNPs’ pores. The analysis revealed carrier stability on no light incidence, while on UV light irradiation, a prompt release was observed. The pores were closed again at a high temperature but tuned for a possible re-opening on being irradiated [146]. Another recent attempt used Janus Au nanostar MSNPs as NIR-responsive carriers. The photolabile molecule split apart on being exposed to NIR radiation, forming succinic acid, which opened the gate channel to assist the drug release [109]. Another eminent effort by Zhao and team demonstrated the visible light-responsive anti-inflammatory essence of azobenzene-modified biodegradable MSNPs and β-cyclodextrin-modified poly(2-methacryloyloxyethyl phosphorylcholine)bMSNPs-AZO/CD-PMPC. Energizing with visible light catalyzed the azobenzene isomerization, which aided the drug release amidst the dermal tissue tracking. The CD-PMPC hydration layer on the NPs’ surface provided lubrication that relieved osteoarthritis [147].

(ii)Temperature stimuli-responsive system

One of the well-demonstrated exogenous stimuli for drug release from MSNPs involves the use of heat from an external agency. This strategy works via thermal oscillation induction in the encapsulated drug molecules. Some studies, however, have also illustrated a synergistic response of heat and chemotherapy, lowering the IC_50_ and moderating the toxic side effects. A noteworthy attempt from 2017 demonstrated an enhanced flexibility of the lipid bilayer and its subsequent liquid state conversion on gaining external heat, improving the tumor cell internalization of DOX [148]. In a relatively uncommon observation, Castillo and group correlated the troubled pathology (tumors, inflammatory or oxidative stress) on 4 to 5 °C enhanced localized temperatures [149]. A unanimous observation in heat-assisted tumor therapies has been the retainment of drugs in the carriers in a normal physiological state followed by the release in the enhanced temperature environment of diseased tissue [150]. Another diligent effort in this direction by Peralta and associates described the thermoresponsive polymer-grafted magnetic MSNPs for targeted drug delivery, noticing a 100% release at 40 °C rather than 25 °C [151]. Similar efforts were made by Shi and group, wherein microwave-driven MSNPs were tuned for drug release via surface-coated thermosensitive peptides [152].

(iii)Ultrasound stimuli-responsive system

Ultrasound exhibits fascinating attributes, such as non-ionization, high tissue penetrability, non-invasiveness, economy and spatio-temporal functions for biomedical applications [153]. This stimulus is regarded as an excellent modulator for drug delivery, owing to its non-invasive penetration deep inside the living tissue. The mechanical, thermal and chemical effects of ultrasound could engineer the drug release from several NCs, such as micelles, liposomes and polymeric NPs [154]. Ultrasound-aided drug delivery from MSNPs has been reported with a copolymer-functionalized surface, capable of gatekeeping abilities across the pores. The copolymer changes from a hydrophobic to a hydrophilic sensitivity upon ultrasound irradiation, inducing gate opening and the subsequent drug cargo internalization [155,156]. To restrict the ultrasound impact to localized tumor tissues, Cheng and colleagues developed an MRgHIFU (magnetic resonance-guided high-intensity focused ultrasound), which sustained the drug release. To monitor the release in vitro, Magnetic Resonance Imaging with gadopentetate dimeglumine (Gd(DTPA)2-) (an FDA-approved gadolinium-based contrast agent) was used. Taking the benefit of MRgHIFU, 3Dimaging and targeting abilities, HIFU-stimulated Gd(DTPA)2- release was directed at the focal point, in an in vivo-mimicking gel phantom. The inspection revealed HIFU stimulation, duration and intensity controlled Gd(DTPA)2- release, signifying the energizing influence of the ultrasound [157]. In a recent study, Amin and colleagues designed a stimuli-responsive DOX delivery system using perfluoropentane (PFP) as a stimuli-responsive agent, ultrasound as the stimulus and lipid-coated MSNPs as the carriers. This system exhibited a localized cytotoxicity induction with enhanced cellular uptake and was determined to be efficient for delivering multiple CDs [158]. 

### 4.2. Dual Stimuli-Responsive System 

This approach uses two external stimuli to streamline the targeted release of moderate drug dosages with enhanced tumor cell internalization. The most common stimuli provided together include the heat and X-ray irradiation. While X-rays engineer the tumor cell internalization, energizing with heat stimulates the release kinetics, ensuring a synergistic influence on cytotoxicity [159]. In a significant 2018 study, Huang and colleagues attempted DOX loading on the hollow HA-functionalized MSNPs (HMSNPs) via disulfide linkages. The HA served the role of capping and targeting agents, while the disulfide linkages conferred a redox-sensitive functioning to the drug-loaded NC. The therapeutic inspection of the NC revealed a sustained DOX release from the HMSNP–SS–HA environment. The results were reciprocated in BALB/c mice with4T1 implanted tumors, exhibiting suppressed tumor growth [160]. A subsequent distinction from Lu and colleagues reported DOX loading (1060 mg·g^−1^) on the α-cyclodextrin-functionalized HMSNPs. The NC exhibited a pH and dithiothreitol concentration-dependent DOX release, evaluated at pH 7, 5 and 4.4, enabling a 58% release at pH 5, which increased to 61% at pH 4.4. An acidic pH dissociated α-cyclodextrin from the protonated aniline, releasing the trapped DOX from the NC pore channels [161]. A more recent study by Song and colleagues used SBA-15 functionalized with Ag NPs before coating with polydopamine (PDA) as an NC to augment curcumin (CURC) release. The binding of CURC with the carrier was controlled via non-covalent interactions with PDA, moderated via pH and oxidative status changes to release the CURC from a CURC–MSNP–PDA–Ag composite. The inspection revealed a site-specific activity in human cervical cancer cells (HeLa) and taxol-resistant non-small-cell lung cancer cells (A549/TAX) [162]. Thereby, the Ag NP–CURC co-delivery through PDA-functionalized MSNPs was optimized by varying the pH and redox status, reducing the undesired spillage under physiological conditions. 

### 4.3. Multiple Stimuli-Responsive System

Multiple stimuli-responsive MSNPs are characterized by assisting the drug release via varying more than two external conditions. For instance, many studies have demonstrated the simultaneous influence of pH, temperature, localized redox status and irradiation on the drug release potential of MSNP cargos. A noted attempt in this regard by Zhang and colleagues used disulfide linkage-responsive, multiwalled carbon nanotubes (MWCNTs) shielded by mesoporous silica-grafted poly(N-isopropylacrylamide-block-poly(2-(4-formylbenzoyloxy) ethyl methacrylate) as a DOX carrier. The DOX remained bound to the NC via covalent binding and physical entrapment, whichvaried with the changing pH, temperature and localized redox environment. The copolymer-stabilized assembly thereby optimized the DOX release, functioning as a need-based gatekeeper [163]. A recent comprehensive review article by Salve and team provides a rigorous description of feasible stimuli to accomplish a sustained drug release with MSNPs. The scientists presented a distinct consideration of MSNPs in light of their high drug loading, tunable pore sizes, ease of functionalization at the internal and external surfaces and largeSA. A noted distinction further mentioned is the adaptability of external and in situ stimulating factors, superseding the similar benefits in Au and Ag NPs due to the porous sensitivity and core–shell architecture of MSNPs [164].

## 5. Breast Cancer: Pathophysiology, Mortality and Chemoresistance

Prevailing as the most threatening cancer affecting females, BC has been a major health concern for more than a decade now, with 2012 GLOBOCAN statistics registering a staggering 18% rise in casualties compared to the number of 2008 cases [165,166]. A predictive analysis by the American Cancer Society anticipates one out of every eight United States females to develop BC in her lifetime. On a global platform, too, trends reflect a significant concern, with an addition of 3.2 million new cases per year until 2050. As rising mortality concerns continue unabated, it is a priority to screen accurate and reliable biomarkers, in addition to timely diagnosis, to improve the success of preventive treatment (Figure 12). Development in diagnostic assays in the past few years has simplified the treatment rigor, whereby rising mortality has witnessed a substantial arrest. Global awareness campaigns and recent research findings have enhanced the unanimity for BC classification, vis-à-vis characteristic histology and molecular events. 

Familiar histological subtypes identified to date include medullary, metaplastic, apocrine, mucinous, cribriform, tubular, neuroendocrine, classic lobular and pleomorphic lobular carcinomas, in addition to the non-specific invasive ductal regimes that characterize the majority of recently screened cases. From the molecular viewpoint, the classification is made via the expression of distinctive surface receptors, categorized as estrogen/progesterone (ER/PR)-positive, negative and triple-negative (no ER/PR and HER2). The ER-positive tumors are rather more common, with a smaller size, lower grading and being lymph node-negative. In a nutshell, Luminal A and Luminal B constitute the ER/PR-positive subgroups, while the ER/PR-negative tumors are distinguished as HER2 (Human Epidermal Growth Factor2), basal and normal subtypes [3,4]. Amongst all the subtypes, the triple-negative (TNBC) state (no ER, PER and HER2) exhibits the most aggressive pathology, contributing to ~10–20% of BC deaths. The lack of positivity of the surface markers renders the TNBC non-treatable with targeted therapies. 

The therapeutic success of tumor treatment is substantially affected by its discrete recognition, accomplished via characteristic OSRs, a summary of which is described in Table 1. For instance, the HER2-overexpressing BC has a higher probability of being implicated in inflammatory responses, with prominent involvement in angiogenesis, lymphangiogenesis and aromatase upregulation. Similarly, toll-like receptors (TLRs) are firmly associated with adhesion and invasion via modulating αvβ3 integrin expression, attenuated TLR-4 activity resulting in reduced IL-6, 8 actions, alongside the TLR-3-mediated inhibition of cell proliferation and survival. These actions of overexpressed surface receptors could be the basis of therapeutic mechanisms in targeted therapies, moderating the stress of excessive CD intake via exercising the local therapeutics on improved tumor cell internalization.

A thorough understanding of overexpressed surface receptor chemistry vis-à-vis ligand specificity could enhance the success of drug delivery to BCs, wherein the advanced stages are often characterized by metastasis to lungs and the brain. Ligand variation is indeed a challenging prospect in this domain, as stoichiometry concerning the optimum drug extent for therapeutic success is a daunting challenge considering the troubled pathophysiology in the tumor vicinity. The versatile features of MSNPs with tailored surface chemistry, varying pore sizes, functionalization diversity and morphology-tuned drug adsorption make them unique and capable of tuning receptor–drug interactions and enhancing tumor cell internalization.

## 6. Discussion of Recent Attempts Using Mesoporous Silica Nanoparticles for Breast Cancer Treatment

This section discusses the major findings of recent studies (2017 onwards) using MSNPs for BC treatment, distinguished via singular and dual therapeutic agents. Emphasis has been laid on comparative IC_50_ reductions, moderated drug release kinetics and enhanced tumor cell internalization abilities. The advantage of dual therapeutic agents is exhibited by the stoichiometric variation of a combinatorial mode, involving the structural changes that decrease the chances of obtaining a resistant response. Although MSNPs do prevail as a carrier in both modes, the therapeutic efficacy of a combined mode is generally higher (likely due to a synergistic influence), owing to which substantial IC_50_ reductions could be accomplished.

### 6.1. Singular Drug Delivery Using Mesoporous Silica Nanoparticles

Commencing from 2017, the first major effort used polydopamine (PDA) and polyethylene glycol (PEG)-coated MSNPs for DOX delivery to MCF-7 and MDA-MB-231 BC cells. The PDA coating enabled a pH-mediated DOX release, while the PEG functionalization conferred an improved physiological biocompatibility. Dispersion profile analysis for the MSNPs in neat, DOX-loaded, PDA and PEG coated states, revealed increments and decrements in PS and PVs. Though PDA and PEG-coated MSNPs exhibited pore sizes of ~200 nm, which were smaller than the cut-off limits of tumor neovasculature pores and exhibited suitability for the EPR effect. Anionic surface sensitivity guarded against reticuloendothelial clearance, facilitating an elongated physiological residence. The inspection of the morphology using Transmission Electron Microscopy (TEM) implied a spherical morphology with porous surfaces for all states of the MSNPs, conveying the structural compatibility of PDA and PEG coating. These physicochemical features together contributed to a ~95% DOX encapsulation efficacy (EE). An analysis via confocal microscopy and flow cytometry for cellular uptake distinctions revealed enhanced extents for PDA and PEG-coated, DOX-loaded MSNPs compared to free DOX. These observations were supported by the MTT assay, which implied higher cell growth and proliferation inhibition for PDA and PEG-coated, DOX-loaded MSNPs than free DOX and the NPs without PEG coating. The antitumor mechanism of intact MSNPs was screened using fluorescence analysis, wherein MSNP–DOX–PDA–PEG-treated MCF-7 cells formed greater autophagic vesicles (green fluorescence) over other configurations. The regulatory influence of the autophagic response was screened via assessing the AKT, mTOR and p70S6K phosphorylations, with MSNP–DOX–PDA–PEG causing the maximum inhibition. The observations were replicated in the nude mice harboring subcutaneous MCF-7 tumors, with MSNPs–DOX–PDA–PEG exhibiting the maximum inhibition in terms of maximum tumor weight decrements. Furthermore, no significant distinctions in the body weight and histopathological abnormalities (heart, liver, spleen, kidney and lungs) for all the carrier configurations imply the biocompatibility of MSNP–DOX–PDA–PEG [169]. This study, therefore, optimized the pH-specific DOX release kinetics via PDA and PEG–MSNP functionalization, conferring high tolerability with enhanced tumor cell uptake.

A subsequent effort in 2017 used folic acid (FA) and N-acetyl glucosamine (NAG)-functionalized MSNPs for DOX delivery to MCF-7 and MDA-MB-231 human BC cells. With a 78 nm PS and a cationic surface sensitivity (unlike the previous study), the prepared MSNPs exhibited compatibility for EPR-driven tumor cell internalization aided by electrostatic receptivity with negatively charged membrane lipids. Monodispersed states with the spherical morphology of prepared MSNPs were inferred by Scanning Electron Microscopy (SEM) analysis, while TEM screening implied their sieve-resembling structure, harboring nanoporous channels on the surface. No structural abnormality was observed for FA and NAG functionalizations in the FTIR screening, but a thermal stability inspection using thermogravimetric inspection revealed a 10% and 19% higher weight loss for NAG and FA-functionalized MSNPs compared to their neat state. This weight loss was attributed to the moderated NAG and FA contributions, with nearly twice the FA molecular weight contributing to its higher loss. Cellular uptake studies using confocal microscopy and fluorescence spectroscopy (comparative red and green intensities) both implied the significant uptake of NAG and FA-functionalized, DOX-loaded nanocarriers (NCs). Quantitative screening by flow cytometry revealed 52.36 ± 3.98 and 60.1 ± 9.37% internalizations for FA and NAG-functionalized NPs in MCF-7 cells, while similar amounts for MDA-MB-231 cells were 52.74 ± 1.08 and 60.58 ± 6.03%. The ligand conjugation of DOX-loaded NCs enhanced their toxicity, as screened by the MTT assay, with 48 and 72 h viability for 100 μg·mL^−1^ extents being 10.68 ± 0.59 and 4.34 ± 0.45% for DOX–NAG–MSNPs and 14.57 ± 0.65 and 8.95 ± 1.61% for DOX–FA–MSNPs. Similar extents for more aggressive MDA-MB-231 cells were 5.41 ± 0.12 and 5.15 ± 0.41% for FA-functionalized and 7.90 ± 1.67 and 5.76 ± 0.31% for NAG-functionalized DOX-loaded NPs. The NAG functionalization exhibited greater cytotoxicity (IC_50_ = 0.86 μM) than FA (IC_50_ = 2.5 μM), which was explained by investigators to be due to a comparatively higher glucose transporter (GLUT) expression on BC cells’ surface compared to the folate receptor α (FRα). The mechanism of the anticancer response was screened using AO/EB fluorescent staining, with minimal necrotic cells exhibiting varied orange-red EB fluorescence, while AO penetrated the intact membrane cells and developed green fluorescence. These distinctions implied apoptosis to be the likely source of cell death following exposure to drug-loaded MSNPs [170].

#### 6.1.1. Distinguished Research Attempts from 2018

Moving to 2018, the first major effort used the YSAYPDSVPMMSK (YSA) peptide sequence to functionalize the MSNPs for DOX delivery to EphA2-expressing SKBR3 (weaker), MDA-MB-231 (moderate) and MCF-7 (strong EphA2 expression) BC cells. The physicochemical and morphological characterization of as-synthesized MSNPs revealed a uniform PS of ~50 nm with a moderately rough surface and a porous texture. The NPs in neat, acid-functionalized and YSA-functionalized states exhibited an anionic sensitivity with the least magnitude of ~13 mV for YSA modification. Entrapment optimization revealed the anionic sensitivity of MSNPs as a facilitator for DOX penetration inside the pores and the concomitant enhanced loading extent. The respective DOX entrapment for MSN and MSN–YSA was evaluated as ~93 and 91%, respectively, signifying a YSA-moderated negative charge on the MSNPs’ surface. Uptake studies using fluorescence spectroscopy revealed a maximum internalization for MSNP–YSA–DOX in the MCF-7 cells due to their higher EphA2 surface expression, but no significant differences were noted for MSN–DOX uptake. The targeting effect of the NC was reciprocated in the cytotoxicity screening, with unaided DOX delivery accounting for the least viability after 12 h treatment at 0, 10, 20, 30 and 40 μg·mL^−1^ DOX content. Surprisingly, on 24 h completion, the maximum viability was noted for free DOX, while MSN–YSA–DOX exhibited the least viability in MCF-7 cells. The effects were not noticed in EphA2-non-expressing MDA-MB-231 and SKBR3 BC cells. These observations were complemented by the insignificant apoptotic distinctions in SKBR3 cells following treatment with free DOX, MSN–DOX and MSN–YS–DOX upon 24 h monitoring. However, the MCF-7 and MDA-MB-231 cells responded differently to all the DOX configurations, probably due to the differential EphA2 surface expression [171].

The second 2018 study focused on CURC delivery to MDA-MB-231 BC cells in xenograft mice models via encapsulation in polyethyleneimine–folic acid (PEI–FA) and hyaluronic acid (HA)-functionalized MSNPs. The characterization of ligand-functionalized NPs implied both PEI–FA and HA to be surface-conjugated via amidation. A morphology analysis (via SEM) of the NPs implied them to be spherical and porous, with 2.665 nm as the average pore diameter, and 1.842 cm^3^ per g as the PV. Interestingly, the surface sensitivity of PEI–FA-functionalized NPs was cationic compared to that of HA-functionalized NPs. Both the ligands enabled a 96% CURC release in the presence of 10 mM GSH, compared to 49 and 48% without GSH. These distinctions suggested gatekeeper-equivalent actions of PEI–FA and HA that mediated a sustained release of CURC and prohibited its abrupt leakage in a non-reducing environment. Screening the cytotoxic effects, the effects on the cell cycle were ascertained following 24 h treatment. The analysis revealed the IC_50_contentsof 37, 27, 22, 14 and 16 μg·mL^−1^ for unaided CURC, CURC loaded on MSNPs, PEI, PEI–FA and HA-functionalized NC configurations, respectively. A lower IC_50_ for CURC loaded on MSNPs than its unaided form suggested structural protection and a higher toxicity for CURC-loaded MSNPs for 20–60 μg·mL^−1^. This extent was further reduced for PEI-functionalized NCs as the cationic PEI ligand bound the CURC more efficiently to sustain its release. The higher cytotoxicity of targeted NC-delivered CURC over other configurations was evidently due to its tumor cell surface receptor and greater endocytosis-mediated tumor cell internalizations. The uptake studies using fluorescence spectroscopy revealed decreasing contents for PEI–FA, HA, PEI and non-functionalized MSNPs, suggesting a greater tumor cell penetration of FA-conjugated CURC-loaded MSNPs. The trends were justified by the investigators in terms of PEI prevalence on the surface of MDA-MB-231 BC cells. These observations were complemented by the greater cell cycle inhibition for PEI–FA-functionalized, CURC-loaded NPs, wherein all configurations exerted an inhibitory response at theG2/M phase. The biocompatibility of CURC-loaded NCs (screened in healthy female KM mice) was inferred by <5% hemolysis and unaltered heart, liver, spleen, lung and kidney functions. All observations were complemented amidst the in vivo antitumor analysis, wherein mice treated with FA-functionalized, CURC-loaded NCs exhibited significantly smaller tumors than those without free CURC. The maximal growth inhibition for tumors was noted for CURC-loaded and PEI–FA-functionalized MSNPs, suggesting the MSNPs’ suitability as a carrier [172].

The next attempt towards MSNP-mediated drug delivery to BC cells used trastuzumab loaded on technetium 99 (TCM-99, magnetically sensitive radionuclide), which constituted the core for prompt BC detection. The physicochemical inspection of the trastuzumab-doped MSNPs revealed a mesoporous behavior (via Powder X-ray diffraction) and TEM implied a PS of 58.9 ± 8.1 nm. Other prominent characteristics included a specific SA of 872 m^2^ per g, a 0.85 cm^3^ PV, a 3.15 nm average pore diameter and a superconducting quantum interference device (SQUID), which suggested magnetic sensitivity with 1.6 emu·g^−1^ saturated magnetizations. The doping stability of trastuzumab was screened using centrifugation, wherein ~99% of the drug content precipitated, suggesting a stable adhesion. The trends were supported by a ~98% entrapment efficacy, and TEM implied a porous texture, with discrete adhesion sites on the surface. The magnetic sensitivity of the core was also ascertained using Magnetic Force Microscopy (MFM), revealing a strong residual effect until an elevation of 50 nm. Low TCM-99 inclusion generated an insignificant magnetic contrast at >50 nm elevation, suggesting a superparamagnetic sensitivity of trastuzumab-loaded MSNPs (Tr-MSNPs). The biodistribution studies of drug-loaded and TMC-99-labeled MSNPs revealed its 30.2% uptake by the lungs, 18.71% by the kidneys (renal clearance), and 4.57 and 1.76% by the blood and heart, implying a long circulation time. The 7.5% Tr-MSNP uptake by the BT-474 tumor cells implied an EPR suitability for NCs, aiding the smart imaging. The pharmacokinetic inspection of Tr-MSNPs in the BT-474 BC cells revealed a 97.37% prevalence in the tumor cells post 2 h injection for a maximum intake. The uptake mechanism involved an initial proximity of Tr-MSNPs to the cell membrane, resulting in van der Waals and covalent interactions between the cell and NPs due to loaded trastuzumab on the MSNPs surface and mesopores. Interactions of NCs with tumor cells resulted in cell membrane invagination, causing its pinch-off and the subsequent generation of endocytic vesicles for trafficking Tr-MSNPs within the cytoplasm. Site-specific Tr-MSNP activities were implied by micro single-photon emission computed tomography (Micro-SPECT), distributing in axial, sagittal and coronal regions (in mice), 30 min, and 2 and 6 h after injection. The EPR efficacy of MSNPs prevented their non-random absorption by other organs, suggesting a stable nanoagent and tumor correlation, in addition to a synergism with therapeutic radionuclides. A cytotoxicity analysis validated the imaging utility of Tr-MSNPs, as the trastuzumab loading extent was low enough and did not cause a therapeutic efficacy, with an IC_50_ of ~830 μg [173].

A subsequent study from 2018 prepared Y_2.99_Pr_0.01_Al_5_O_12_-based (YP) mesoporous silica-coated NPs, which were then functionalized with protoporphyrin (PpIX) and FA, to retrieve a YPMS–PpIX–FA nanocomposite that was screened for deep photodynamic therapy (PDT). The investigators’ intent for this nanophosphor (NPR) development was to supersede the inadequate luminescence of scintillating NPs that weaken the energy transfer to the photosensitizer. The sol–gel-method-prepared NPR was optimized for transducing the X-rays’ penetrating ability by covering with mesoporous silica. The YP selection was optimized via its luminescence emission within 300–450 nm, which could activate the photosensitizer, PpIX. The coating of the mesoporous silica shell on the YP core surface was screened via TEM, while the retainment of crystallinity by the YP core after silica coating was inferred from X-ray diffraction (XRD). To optimize the porous silica periphery for biomedical applications, multiple concentrations of TEOS were used and, with an extent >0.1% (*v*/*v*), 20–50 nm shells were retrieved with a self-nucleated state of silica. The optimized complexes were thoroughly screened by TEM, DLS and BET, which unanimously suggested their nanoscale functionalities with a 70 m^2^·g^−1^ SA of silica layer compared to 7 m^2^·g^−1^ for yttrium phosphide (YP) NPs, forming a porous layer. The luminescence screening of the optimized NPR suggested adequate PpIX and nano-scintillator overlap, with which the emission spectra agreed (with and without PpIX), indicating an efficient energy transfer for deep PDT. These predictions were reciprocated in the in vitro studies, with the preferential uptake of YP nanocomposites in folate receptor-expressing 4T1 BC cells. The conjugation of PpIX with UVA in the in vitro models suggested the generation of ROS, killing the tumor cells, with no loss of PpIX activity on its nanocomposite conjugation. The effects were also replicated in the in vivo (male CD-1 mice with implanted tumors), with no random toxicity and histopathological abnormalities in the kidney, liver, or lung tissue sections of treated mice [174]. This investigation illustrated the unique potential of a YP–mesoporous silica combination to attain a localized photothermal effect for deep-seated tumor treatment. A valuable insight from this study invites a possible study of such complexes with moderated CD extents.

The last of the 2018 attempts grafted hyperbranched polyamidoamine (PAMAM) as a functionalizing moiety for amorphous silica NPs, followed by their conjugation with fluorescent dyes and covalent attachment with anti-HER2 antibodies. The TEM inspection of polyamidoamine-coated silica NPs (PCSNs) revealed a 40 nm PS, with a porous texture within the SKBR3 tumor cells, alongside an occasional prevalence within the lysosomes. The tumor-specific uptake of PCSNs was screened by in vitro fluorescence sensitivity, with the Alexa 488 fluorescent-tagged PCSNs binding to the SKBR3 tumor cells in the initial 4 h, becoming internalized upon24 h incubation in the culture medium with 1200 ppm PCSNs. The tumor cell specificity of PCSNs was ascertained using confocal microscopy, revealing an overlap of red, lysosomal and green PCSN probes. To ascertain the therapeutic efficacy of dye-conjugated and anti-HER2 Ab-conjugated silica NPs, the polyamidoamine-coated silica NPs (PCSNs) were incubated with HER2-overexpressing SKBR3 BC cells prior to X-ray irradiation. The cell proliferation studies revealed no significant effect on 24 h viability at <1200 ppm PCSN, and the noticeable inhibition of cell viability occurred only at 2400 ppm. The synergism of PCSNs with radiation efficacy revealed a higher inhibition of SKBR3 cells for 6, 600 and 1200 ppm PSCNs with 8 Gray (Gy, measuring unit of radiation intensity) radiation compared to the individual PCSN or X-ray irradiation. The corresponding effects on apoptosis were monitored using terminal deoxynucleotidyl transferase-mediated nick end-labeling (TUNEL) and fluorescent-labeled inhibitor of caspase (FLICA) assays. As the PCSN probes were first internalized and then irradiated by X-rays, they could be considered as localized sensitization-mediated outcomes in HER2-overexpressing BC cells [175]. Therefore, this study provided a novel interface for tumor cell-specific therapeutics by irradiating the tumor-residing MSNPs.

#### 6.1.2. Salient Attempts from 2019

The first 2019 study focused on energizing the transport of pepstatin A (a peptide that is a strong cathepsin D, an aspartic protease overexpressed in BC) across the cell membrane. The investigation screened MSNPs with large pores (LPMSNs) and those with hollow organosilica (HOSNPs), both prepared using the sol–gel process. To screen the therapeutic efficacy of NCs, both configurations of MSNPs were loaded with pepstatin A before being incubated with MCF-7 human BC cells. The LPMSNs and HOSNPs both exhibited their nanoscale attributes reasonably well with TEM-screened particle diameters (PDs) of 100 nm. The specific SA of 817 m^2^·g^−1^ with 3–7 nm pore sizes (for LPMSNs) and 58 m^2^·g^−1^ with a 26 nm pore size (for HOSNPs) were the other prominent nanoscale traits. A major distinction was the development of negative (−19.5 mV) and positive (32 mV) surface polarities for LPMSNs and HOSNPs at 7.4 pH. More interestingly, the loading with pepstatin A decreased the LPMSNs’ *ζ*-potential from −19.5 to −22.6 mV, while that of HOSNPs decreased from 32 mV to −13.2 mV, suggesting an entropically triggered reorientation. The distinctions were correlated with the surface adsorption of pepstatin A (in HOSNPs), while in LPMSNs, it adsorbed into the pores. In terms of therapeutic efficacy, the LPMSNs exhibited a 20% tumor cell death, which was surprisingly 60% for HOSNPs. Such distinctions were uncharacteristic, as LPMSNs possessed a higher loading and were countered by the possibility of premature release (before endocytosis) while being incubated with tumor cells. Contrary to this, the HOSNPs bound anionic pepstatin A to a stronger extent due to electrostatic interactions with cationic HOSNPs’ ammonium moieties. These distinctions contributed to a sustained release and a higher cell death for tumor cells treated with HOSNPs [176]. Therefore, this study demonstrated the distinct therapeutic efficacies for hollow and solid MSNPs, with the former ensuring a gradual release and minimized instantaneous patient sensitization.

A subsequent 2019 study investigated the ability of MSNPs to deliver DOX in the MCF-7 human BC cells overexpressing the gonadotropin-releasing hormone (GnRH). The optimized configuration of MSNPs (conjugated with decapeptide) was functionalized with GnRH using PEG as a linker molecule. The dispersion screening of as-synthesized MSNPs corroborated well their nanoscale features, with TEM implying a hydrodynamic size (HS) of 55 nm and −24.39 ± 3.2 mV as the *ζ*-potential. The PEG–triptorelin conjugation on the surface of the MSNPs increased the HDS to 159 ± 50 nm and the *ζ*-potential to −17.94 ± 1.89 mV. The morphology screening suggested a porous texture, with SEM implying a spherical texture. No compositional abnormality was screened for the functional MSNPs during FTIR, except for the C = O and N-H functionalities. The DOX loading on targeted MSNPs was done via stirring DOX and MSNPs in a 1:1 weight proportion. A loading extent of ~52% was computed for the targeted state of MSNPs, which was ascertained in the broad peak of the increased intensity of –OH groups at 3550 cm^−1^. Monitoring the release kinetics, pH 5 exhibited a 96.4% DOX discharge in 96 h compared to the 49.52% at a pH of 7.4. The release profile of MSNP–PEG–triptorelin (a GDH agonist) followed an initial burst regime with a subsequently sustained profile. The cellular internalization of targeted and neat MSNPs was monitored using confocal microscopy, wherein a higher tumor cell residence was noted for targeted MSNPs, quantified via the internalized DOX intensity using confocal microscopy. The cytotoxicity screening using MTT assay revealed significant growth inhibition with 10 μM DOX in the targeted delivery mode over 24 h, complemented by the least viability of 34.6 ± 2.76% compared to 54.5 ± 2.8 and 46.8 ± 1.82% for free DOX and DOX-loaded, non-targeted MSNPs. The observations met support from the 0.44 and 0.65 μM IC_50_ extents for targeted and non-targeted MSNPs, respectively. The preferential actions of DOX-loaded, targeted MSNPs in MCF-7 cells were screened using AO/EB double staining, with a maximum green intensity for the MCF-7 cells, signifying a site-specific action of the targeted MSNPs. Few tumor cells exhibited an orange pattern, suggesting the manifestation of early apoptosis. The quantification of these observations via Annexin V-FITC staining amidst flow cytometry revealed 36% apoptosis compared to the 13% for non-targeted ones [177]. This study, therefore, predicted the ligand suitability of GnRH for targeted delivery in hormone-sensitive BC cells.

#### 6.1.3. Major Research Attempts from 2020

Amongst the 2020 studies focusing on BC treatment using MSNP-mediated therapies, the first attempt focused on umbelliferone (UMB, a natural coumarin derivative) delivery to MCF-7 human BC cells, via loading on the MSNPs’ pores. The monodispersed state of loaded UMB with reduced self-interactions was ensured via capping with pH-sensitive polyacrylic acid (PAA). The PAA-capped and UMB-loaded nanohybrid was surface-conjugated with FA. The structural intactness of the drug-loaded NCs was screened using Energy Dispersive X-ray (EDX), XRD and FTIR spectroscopy, wherein no abnormality and successful functionalization of the MSNPs’ surface were inferred. An analysis via TEM revealed a spherical morphology with an average diameter of 40–50 nm, 238.85 m^2^·g^−1^ specific SA and 0.514 cm^3^·g^−1^ cumulative PV, inferring the nanoscale attributes of NH_2_-functionalized MSNPs. However, upon UMB loading and PAA grafting, the specific SA and PV reduced to 8.206 m^2^·g^−1^ and 0.058 cm^3^·g^−1^, respectively, reflecting the UMB–NC interactions. The physicochemical inspection of UMB-loaded NCs revealed a 12.56% loading efficacy with anionic sensitivity (due to PAA capping); albeit in the blank state, the surface receptivity was cationic. The stability of UMB-loaded NCs was monitored in an in vivo-mimicking 10% FBS aq environment, wherein no drastic changes in HDS were noticed until7 and 14 days (414.1 ± 20.9 to 421 ± 5.3 nm). An unchanged fluorescence on UMB loading suggested its efficacy and compatibility with the native NC structure. 

The drug release efficacy of UMB-loaded NCs was monitored at pH 5, 6 and 7.4, with the maximum release of ~52% being noticed at pH 5. The therapeutic screening of drug-loaded NCs was assessed in vitro by the significant oxidative stress and mitochondrial damage in theMCF-7 BC cells overexpressing the folate receptor. No such observations were noticed for unaided UMB administration. The results were also replicated in vivo in the xenografted mice tumors, with the MSNP-delivered UMB causing a significant reduction in tumor growth due to its sustained release and homogeneous biodistribution. The specific localization of UMB-loaded NCs in the tumor cells was inferred via their 2.5 and 7-fold enhanced intracellular UMB secretions in the MCF-7 cells for PAA-capped and FA-targeted NCs [178]. Thus, this study illustrated the influence of ligand-targeting mediated-improved UMB therapeutic efficacy, characterized by structurally guarding nanosized pores and a sustained release profile. Figure 13 depicts the chemical structures of UMB, coumarin and PAA, wherein UMB with an additional –OH group probably seems more water-soluble. Having PAA as a capping agent is a good argument for minimizing UMB–UMB interactions, which are progressively replaced by non-covalent UMB–PAA receptivity. Using PAA as capping agent infers a likability vis-à-vis philicphobic equilibration and a weak dissociation at physiological pH, making them suitable for ensuring the uniform distribution of bioactive compounds. Furthermore, PAA use is a good argument for its evaluation with potent anti-oxidative heterocyclic compounds, having UMB-resembling structural aspects. 

A subsequent study used transferrin-conjugated MSNPs as therapeutic agents for α-therapy and radionuclide delivery to BT-549 BC cells. The choice of transferrin as a ligand was due to its propensity for multiple cancer cells, further engineering its functionalization with 3,4,3-LI(1,2-HOPO), a chelator with a high binding affinity for f-block elements. The optimized MSNPs exhibited a hierarchical structure with 3.2 ± 0.6 nm as the pore diameter and 23° as the characteristic XRD peak. To ensure long-term stability, the designed NPs were grafted with β-cyclodextrin. 3,4,3-LI(1,2-HOPO) and this was inferred via an EDX-characterized surface organic layer, increasing the hydrodynamic diameter from 152 ± 10 to 173 ± 6 nm. These features served as the basis for screening the tumor cell internalization of ^225^Ac (actinium) and ^238^Pu (polonium), clinically validated radioisotopes. The treatment at 100 μg·mL^−1^ (with and without f-block elements) for 24 and 48 h decreased the tumor cell viability to a greater extent with ^225^Ac–HOPO–MSNPs. Similarly, treatment with 1000 μg·mL^−1^ for 48 h decreased the cell viability, but the distinctions for those treated with ^225^Ac–HOPO–MSNPs were statistically significant. The tumor cell internalization of radionuclides was screened with confocal microscopy, with insignificant toxicity without transferrin targeting. The biodistribution and excretion of NCs were screened in mice, wherein ^238^Pu-loaded NPs showed higher tracer suitability due to their low activity. Monitoring the intravenously injected radionuclide-conjugated NCs for 48 h revealed a steady body weight with no dermal infection and impaired mobility. The MSNP-mediated ^238^Pu delivery eliminated its 76 ± 3.1% proportion in 48 h compared to the insignificant amount (~32%) for a citrate solution. The residual deposition of radionuclides in the bone matrix was also reduced in their MSNP-mediated delivery, with MSNP-delivered ^238^Pu mostly residing in mice livers upon 48 h completion, determining it to be the likely excretion route for the HOPO (chelator)-conjugated ^238^Pu MSNPs [179]. Thus, this study explored the MSNPs’ suitability as a carrier for the radionuclide delivery to the BC cells. The results offer encouragement for similar activities of radiolabeled isotopes and their stoichiometrically optimized combinations with CDs and multiple nutraceuticals.

An ensuing 2020 attempt using MSNPs for BC drug delivery comprised chemo and immunotherapy in combination for DOX delivery to 4T1 BC cells. The unique aspect of this study involved the capping of MSNPs with the cancer cell membrane, with X-ray and ROS-responsive diselenide bonds accomplishing a steady release of DOX. No targeting ligand was used to modulate the MSNPs’ therapeutics, making the cancer cell membrane coating a new target localization strategy. The coating with a cancer cell membrane improved the uptake of DOX-loaded MSNPs (CM–MSNP–DOX) by the tumor cells, noted as significantly higher for MCF-10A (breast epithelial cells) and RAW264.7 macrophages. These observations were reciprocated in the analysis of the tumor cell growth inhibition, with CM–MSNP–DOX killing a greater number of cells at lower extents. Contrary to this, CM-devoid MSNPs carrying DOX required a much higher extent to achieve an equivalent cell death rate. A combination with 1 Gy X-rays resulted in a 2.1-fold lower IC_50_ of CM–MSNP–DOX than their uncoated configurations. It was also noticed that X-ray irradiation augmented the DOX actions on calreticulin and chromatin-binding high mobility group 1 (HMGB1) protein towards activating the antigen-presenting cells. Contrary to the 6.6 and 5.7 h half-life durations for DOX-loaded MSNPs and unaided DOX, respectively, the CM-coated, DOX-encapsulating MSNPs exhibited a t_1/2_ of 18.4 h, suggesting a stealth delivery. The effects were successfully replicated in 4T1 orthotopic mammary tumor cells in mice, with a higher therapeutic efficacy for a combination of low-dose X-ray irradiation and CM–MSNP–DOX. Encouraged by these observations, co-treatment with programmed death ligand-1 (PD-L1), an immune checkpoint inhibitor, was also explored in vivo, which fared better, with cyclic decrements in mice body weight and enhanced levels of hepatocellular and renal enzymes (aspartate aminotransferase, alanine aminotransferase, urea nitrogen and creatinine). Unfortunately, the animals treated with a CM–MSNP–DOX and PD-L1 combination also exhibited higher cardiovascular toxicity than the singular groups, suggesting a need for dosage monitoring [180]. The capping of NCs with the cancer cell membrane is an uncommon therapeutic strategy and could bypass the need for a stabilizing agent against chemical-environment-driven aggregation.

The last major 2020 research effort exploring the drug trafficking ability of MSNPs used an FA and RGD tripeptide (Arg-Gly-Asp) dual-targeted (MSNP–NH_2_–FA–RGD) carrier to deliver PTXto MCF-7 human BC cells. Similar to several previous attempts, the TEM analysis of the prepared MSNPs revealed a spherical morphology with a smooth surface enriched with ordered mesopores. The assessment of the dispersion characteristics revealed a PS of 204.1 nm for MSNP–NH_2_–FA–RGD, with a PDI of 0.269. Sequential modifications resulted in varying surface sensitivity, with anionic proximity for MSNPs in distilled water (18.4 ± 4.30 mV); however, –NH_2_ functionalization resulted in the reversal of this surface sensitivity (25.6 ± 3.8 mV). Subsequently, PEG long chains enclosing the targeting –NH_2_ moiety of MSNPs moderated the surface charge to 22.9 ± 3.9 mV. The reduced SA from the neat state of MSNPs (1203.41 m^2^·g^−1^) to the sequential conjugated –NH_2_(659.34 m^2^·g^−1^), -NH_2_-FA (514.86 m^2^·g^−1^) and NH_2_–FA–RGD (426.68 m^2^·g^−1^) suggested enhanced nanoscale effects, complemented by 3.77, 3.09, 3.07 and 3.03 nm pore sizes and 1.13, 0.51, 0.40 and 0.40 cm^3^·g^−1^ PVs. The therapeutic significance of PTX-loaded MSNPs revealed a 18.7% loading with an 85.2% EE, suggesting a sustainable release ability of RGD-targeted MSNPs. The 1.6-fold increased IC_50_ extent of MSNP–NH_2_–FA–RGD-conjugated PTX compared to its unaided delivery implied a higher localized toxicity induction by the FA and RGD targeting of the –NH_2_-functionalized MSNPs. The tumor cell specificity of the PTX-loaded NCs was inferred from the confocal microscopy, wherein MSNP–NH_2_–FA–RGD developed a red fluorescence in MCF-7 cells with a higher fluorescence than the MSNPs–NH_2_ and MSNP–NH_2_–FA configurations, suggesting the targeting efficacy of FA coating and RGD peptide conjugation [181]. Though the results offered valuable insights for designing targeted drug carriers, not being conducted in animal models necessitates the in vivo replication of the experiments for a prompt clinical inspection. 

#### 6.1.4. Major Studies Reported in 2021

The year 2021 witnessed a continued persuasion for using varied MSNP-functionalized textures to treat BC, illustrating the robustness of functional activities. In the first major attempt, Day and colleagues prepared the MCM-41 morphology of MSNPs via the sol–gel approach, using CTAB as a template and TEOS as a silica precursor. The prepared MSNPs were screened for their nanoscale attributes, exhibiting a 150 nm diameter (via TEM) and a >1000 m^2^·g^−1^ SA with a long-range patterning of the inner mesopores. These characteristics propelled an EPR suitability of the MSNPs vis-à-vis leaky tumor vasculature. A uniform dispersion was revealed by Dynamic Light Scattering (DLS), with a 232.97 ± 9.56 nm PS and a 0.31 PDI. To optimize the as-formed MSNPs for the site-specific delivery of tamoxifen (TAM, works on overexpressed estrogen receptors in BC), its MSNPs’ linkage was facilitated using poly-L-histidine (PLH) as a pH-sensitive gatekeeper. The PLH functionalization and subsequent TAM conjugation of the therapeutically active NPs was characterized using FTIR, XPS (binding energy) and TGA (thermal stability of conjugations). No structural abnormality was noticed, and a grafting stoichiometry of ~29% was accomplished. No therapeutic studies on the cell lines or animal models were conducted, but the investigators emphasized that the stable PLH–TAM conjugation with a nanoscale architecture had therapeutically significant structural prospects [182].

The next 2021 study used unaltered and chemo-response-tuned small (sMCM-41) and large-sized (lMCM-41) MSNPs for 5-FU delivery to MCF-7 human BC cells. Using CTAB as a template and TEOS as a precursor, the prepared MSNPs exhibited a 1969 m^2^·g^−1^ and 780 m^2^·g^−1^ SA in their unmodified (sMCM-41) and thiol-functionalized states. Studying the tumor cell internalization abilities of thiol-modified MSNPs, the investigators demonstrated the inability of large-sized NPs to move across the deep intracellular compartments, substantially prevailing around the plasma membrane. The sMCM-41 is internalized efficiently within the tumor cells, majorly prevailing in the cytosol and minorly within the nucleus, contrary to the extensive nuclear residence for thiol-functionalized particles. Monitoring the sMCM-41 and lMCM-41 actions on mitochondrial depolarization, the thiol-modified lMCM-41 exhibited significant modulations. The study of the 5-FU release kinetics in the unmodified and thiol-modified carriers revealed a diffusion–erosion mechanism in the absence GSH, with both coefficients being <0.45, following Fickian diffusion. However, with GSH, first-order release sensitivity was noted, the modulated redox status breaking the disulfide linkages and changing the carrier’s functional SA. The distinctive effects on modulated gene expressions implied the capability of unmodified sMCM-41 to alter the PI3K/Akt pathway, FOXO signaling and actin cytoskeletal functioning. The thiol-functionalized sMCM-41, however, induced significant changes in mitochondrial apoptosis, AMPK, FOXO and Wnt signaling, together associated with proliferation, migration and survival. The interactions of lMSM-41 with focal adhesion, adherence pathways and membrane-driven ErbB signaling suggested few alterations of the mitochondrial/nuclear transcription. Screening the 5-FU efficacy in the in vivo conditions, sMCM-41 particles exhibited a higher number of BC-regressing targets, ascertained via the body weights and tumor volumes (TVs) comparisons for 5-FU-loaded, thiol-modified MCM-41-treated nude mice and the control carriers. Mild stroma formation was noticed for the 5-FU-delivering NCs, while it was prominent in the saline-administered animals. Other major distinctions included minor megakaryocyte hyperplasia and lymphoid follicles in the animals treated with thiol-engineered, 5-FU-loaded sMCM-41 NCs [183]. 

The next major attempt towards the utilization of MSNPs as BC drug carriers used them in a hollow architecture, via the biotemplate-directed method. The designed configurations were comprised of: (i) silica nanorods (NRs), (ii) dendritic fibrous nanostructured silica (DFNS) and (iii) ultraporous sponge-resembling DFNS. The preparation of the biotemplate was completed in three steps, the first of which was the fabrication. For this, cellulose nanocrystals (CNCs) containing rod-shaped NPs were synthesized via the acidic hydrolysis of cellulose fibers. Henceforth, the core–shell NPs were made using CNC particles as hard templates, via (i) polycondensation at the CNCs’ surface, or (ii) bicontinuous microemulsion-mediated DFNS synthesis. The patterning of the DFNS on the CNCs’ surface gave core–shell CNC–DFNS structures, after which the CNCs were removed from the core via calcination, to give hollow structures. The CNCs were prepared in uncoated and CTS-coated forms. All structures were thoroughly characterized by FTIR and the stability via TGA, and no impurity was noticed. The effect of TGA was a loss in the functionalizing moiety, a quite familiar aspect of MSNPs. The physicochemical and morphological characterization of as-prepared silica nanoconstructs revealed anionic surface receptivity with a 0.14–0.31 range in PDI and a <600 nm particle diameter (except ultraporous sponge DFNS) (via DLS), 202–736 m^2^·g^−1^ SA, 2–20 nm average pore size and 0.3–2.5 cm^3^·g^−1^ PV (via BET). Coating with CTS enhanced the particle diameter and decreased the SA, pore size and PV, increasing the nanoscale attributes. The pH-optimized DOX loading and in vitro release from the prepared silica nanoconstructs varied with their porosity and dimensions of the hollow cavity, wherein CTS-capped hollow silica NRs exhibited a lower loading capacity than others. Similar effects were noticed for the release, with CTS-capped HSNRs exhibiting a longer initial burst regime until 30% discharge, whereas CTS-capped, hollow DFNS and ultraporous sponge-like DFNS produced a similar effect with 10% drug loss. The shape effect was visible in the CTS–hollow DFNS, with a significant enhancement at 60 h, due to the hollow space in the MSNPs’ cores. Such distinctions could be attained only with adequate pore depletion in the shell, priming major DOX release from the mesopores until 60 h. The comparatively low, post 24 h DOX release from all carrier configurations was due to electrostatic interactions between anionic CTS and cationic DOX. The hollow silica NR morphology was selected (optimum shape and size) for monitoring DOX delivery in the MDA-MB-231 TNBC cell line. Monitoring the tumor cell viability at IC_50_ of DOX (1.26 μM) and its quadruplicate extent (5.04 μM) for 24 h, substantial decrements were noted for CTS–HSNR–DOX, with no HSNR and control distinctions, implying biocompatibility. Similar consequences were noted at a higher extent, except for greater cytotoxicity and anticancer ability [184]. Thus, this comprehensive attempt provided critical insights into the shape-modulated MSNPs’ therapeutic efficacy for their drug loading and release potential.

The next major research attempt from 2021 using MSNP-mediated drug delivery to BC cells comprised a green approach, wherein rice and wheat husk were used to prepare MSNPs. The benign aspect of the adopted methodology was the acid-leaching-driven extraction of biogenic silica from the cereal husk on its sodium silicate (Na_2_SiO_3_) conversion. Henceforth, MSNPs were synthesized via the addition of Na_2_SiO_3_ to the template mixture, in continuous and discrete modes, following the sol–gel process. The structural and morphological inspection of the rice and wheat husk-prepared MSNPs was done using XRD, FTIR, BET and SEM, wherein rice husk nanocarriers (RHNs) exhibited a higher crystallinity than the slit-morphology pores. Contrary to this, wheat husk nanocarriers (WHNs) revealed spherical NPs with narrow cylindrical nanopores as constituents. A striking distinction concerning the precursor addition was noticed, with the discrete inclusion increasing the hydrophilicity, PS and pore size compared to the continuous regime, which had a high precursor concentration. The drug trafficking ability of RHNs and WHNs was screened by monitoring the DOX release at 5.4 and 7.4 pH, where a more acidic environment enabled almost twice the DOX release due to its higher aqueous solubility. Here, again, the role of the shape-affected drug release was noticed, with the discrete-mode NCs resulting in DOX accumulation due to their greater pore diameter. These distinctions were complemented by the cytotoxicity evaluations, wherein RMSN-D and WMSN-D (discretely added) exhibited a pronounced effect compared to other configurations (RSN, WSN, RSN-C and WSN-C). No toxic induction of blank NPs in normal cells implied the NCs’ biocompatibility. The mechanism of anticancer potency was ascertained as apoptotic by screening the fluorescence patterns, wherein cells treated with DOX-loaded RMSN-D and DOX revealed chromatin condensation and fragmentation [185]. Therefore, this study provided a fundamental understanding to tune the hydrophilicity and pore size of MSNPs via discrete or continuous addition of a precursor. 

#### 6.1.5. Significant Research Attempts from 2022

The year 2022 has witnessed significant progress in the utilization of MSNPs for drug delivery to BCs, suggesting continued interest in using the functionalization and shape–size-modulated carrier attributes of MSNPs. The first major attempt is a study by Laranjeiria and colleagues, wherein MSNPs with a magnetic sensitivity were used to deliver the low-bioavailability drug, exemestane. Initially, iron oxide NPs were prepared using FeCl_3_·4H_2_O and FeCl_2_·4H_2_O as precursors, NH_4_OH as a reducing agent and citric acid as a capping agent. The prepared iron oxide NPs were activated on mixing with chloroform and a subsequent sonication for 60 min. Thereafter, CTAB and TEOS were added as templates and precursors for MSNP synthesis. Subjecting this mixture to overnight stirring gave a microemulsion texture, which was evaporated (to eliminate CHCl_3_) and optimized with the addition of CTAB and NH_4_OH. Finally, the mixture was enriched with CTAB again at 80 °C, giving brown colloidal NPs that were separated using centrifugation. The intent of the investigators to confer magnetic sensitivity to MSNPs was to use them for Magnetic Resonance Imaging (MRI), monitoring the tumor development and treatment. The physicochemical characterization of functional IOMSNPs (iron oxide mesoporous silica NPs) revealed 0.224 to betheir PDI (a uniform dispersion) and a PS of 137.2 nm (via DLS), corroborating their nanoscale attributes. Screening via TEM revealed an average size of 90.3 nm, with a spherical morphology surrounded by a dendritic msopore network. Moreover, the analysis also confirmed the core of the particle was constituted of magnetite and silica, with a hexagonal morphology similar to MCM-41. Screening the exemestane delivery potential of particles revealed a loading capacity of 37.7% and an EE of 90%. Sustained binding and release kinetics were inferred as the maximum drug content was released after 24 h, confirming its adsorption within the cores and not on the external surface. Although no in vivo experiments were undertaken, the toxicity of the designed NPs was studied via effects on cell numbers amidst the HGnF cell culturing. A <30% reduction in cell viability implied insignificant toxicity for IOMSNPs, which was also ensured by the retainment of the elongated morphology upon 24 h exposure to cultured HGnF cells. The magnetic sensitivity of as-synthesized IOMSNPs revealed a high saturation magnetization (34.3 emu·g^−1^) with a superparamagnetic behavior, the essential requirement for an efficient MRI agent. The performance of NPs as MRI agents was inferred from the decreased T2-weighted image signal upon increasing the NPs’ concentration, gathering support from earlier studies correlating the superparamagnetic nature for similar outcomes [186]. 

The next study of the year 2022 used 3-carboxy-phenylboronic acid (PBA) as a ligand for PAA-gated MSNP-mediated cuminaldehyde (CUM) delivery to MCF-7 human BC cells. The effects were screened in targeted and non-targeted configurations, in the in vitro and in vivo settings (4T1-induced tumors in female BALB/c mice). Higher therapeutic efficacy was noticed for targeted NCs viaG2/M-phase cell cycle arrest and apoptotic induction via the loss of mitochondrial membrane potential (MMP, intrinsic pathway). The effects were replicated in the in vivo experiments, with noticeable reductions in tumor growth and volume. Uptake studies revealed 1.9 and 4.9-fold higher CUM extents for the PBA-targeted NCs, while a much lower internalization was noticed in non-cancerous NKE cells, due to a lower sialic acid expression, which was much higher on MCF-7 BC cells. Monitoring the pH-varied CUM release, the acidic environment (pH 5) exhibited a 48% liberation upon 48 h monitoring compared to 14 and 6% at pH 6 and 7.4, respectively [187]. Figure 14 depicts the chemical structures of CUM, PBA and PAA, wherein plentiful hydrogen-bonding (HB) sensitivities project a likelihood of pulsating interaction shifts in trafficked CUM. These functional intricacies fit well with the CUM and PBA diffused electron densities, determining their mutual proximity. 

The inclusion of PAA as a capping agent moderates these interactions via PBA engagement. Probably, such an arrangement makes way for moderate CUM binding that, in turn, sustains its release at <7 pH. The prevalence of HB sensitivity in sialic acid seems a supporting factor for PBA binding that mediates a graduated CUM release (Figure 14). No covalent or ionic interactions were noted in this proximity that could have strongly bound CUM and prohibited its tumor cell trafficking vis-à-vis varying chemical environment of the vicinity. A likely interest is created for knowing the drug release at <5 pH, as at pH 5, a higher CUM release was noticed. 

A subsequent 2022 attempt aimed at BC treatment using MSNPs with incorporated guanidinium ionic liquid (GuIL) in the pores and with FA–PEG and PEI polymers as surface-functionalizing agents. The prepared NPs were conjugated with siRNA for its delivery to MDA-MB-231 TNBC cells, optimizing their diameter to internalize the ~2 nm-sized siRNA. Other prominent optimized features included a cationic surface and core sensitivity, maximizing the carrier–tumor cell proximity. The binding controls for siRNA cargo-loaded FA-PEG/PEI-GuIL-KIT-6 comprised anionic siRNA and cationic FA-PEG/PEI-GuIL-KIT-6 electrostatic attractions, decreasing the *ζ*-potential from 39.75 (without siRNA) to 18 mV. The enhanced nanoscale attributes were noticed for siRNA-loaded NCs, with 76.01, 86.73 and 81.54% decrements in pore-specific SA, PVs and pore diameters, respectively, from the KIT-6 stage to the siRNA-loaded, FA–PEG and PEI-functionalized state. Screening the siRNA loading efficacy vis-à-vis pore sizes, a 3.5 nm pore size was observed as the optimum, exhibiting a 96% loading. A 74% in vitro siRNA release in 120 h at 37 °C from the NCs suggested their sustained activity, wherein biocompatibility was monitored via assessing MDA-MB-231 and MCF-10A cells’ response to increasing FA–PEG/PEI–GuIL-KIT-6 extents for 24 and 48 h. No significant viability decrements with NCs implied their compatibility with a tumor cell-specific toxicity due to tagged siRNA. These observations were complemented by uptake studies via confocal laser scanning calorimetry, wherein 63.9% FA–PEG/PEI–GuIL–KIT-6 absorption was noticed in 4 h incubation. Inspection using fluorescence revealed reduced red and enhanced green emission post 6 h, implying an endosomal rupture and concurrent escape from endolysosomes. For a possible EGFR involvement, the analysis of treated and untreated cells revealed 52 and 87% EGFR1 mRNA knockdown at 24 and 48 h, suggesting its therapeutic rationality [188]. Thereby, this study demonstrated the conceptualization of siRNA administration via NCs via EGFR targeting in MDA-MB-231 BC cells. The cationic receptivity of FA and PEI aided the tumor cell uptake of the siRNA-loaded carrier.

Yet another significant attempt from Lee and colleagues used mannose (MAN) and polyacrylic acid (PAA)-anchored, -NH_2_-functionalized MSNPs for DOX delivery to MDA-MB-231 TNBC cells. Coating with mannose significantly improved the DOX toxicity, screened via reduced RBC hemolysis. Screening the morphology and dispersion profile, the NH_2_-functionalized MSNPs exhibited a spherical shape with a uniform distribution and cross-linked MAN–PAA copolymer. The surface polarity shifts comprised anionic (~30 mV) sensitivity for native MSNPs to cationic (~11 mV) for the –NH_2_-functionalized state. The PS, however, increased from 100.38 ± 13.67 nm (for MSNPs) to 134.45 ± 8.30 nm (on –NH_2_ functionalization). Similar to earlier attempts, the SA, pore sizes and PV decreased from the neat to the –NH_2_-functionalized state, suggesting a progressively enhanced nanoscale effect. The release kinetics were studied at pH 4.2, 6.8 and 7.4, with and without MAN, with the highest release of 85.04 (from DOX–MSNPs) and 77.70% (from DOX–MSNP–MAN–PAA) at pH 4.2, post 48 h of delivery. The MAN coating of NCs reduced the DOX toxicity (the RBC hemolysis) from >60% (free DOX) to >20% (for DOX-loaded MSNPs) and merely 3.10% (for PAA–MAN-coated NCs). Coating with MAN moderated the surface silica and membrane protein interactions for an enhanced biocompatibility. These observations found the least IC_50_ extents of MAN–PAA-coated DOX–MSNPs compared with free DOX and DOX–MSNPs. 

Uptake studies using fluorescence microscopy revealed a higher DOX internalization inMDA-MB-231 BC cells than MCF-7 cells, wherein the mean fluorescence intensity (MFI) was 4.25 and 2.17-fold greater than for DOX and DOX–MSNPs. These differences were insignificant for MCF-7 BC cells, suggesting a preferential uptake of MAN–PAA–DOX–MSNPs by MDA-MB-231 TNBC cells. Relying on the reported DOX anticancer mechanism via intracellular ROS enhancement, the MAN–PAA–DOX–MSNPs exhibited 244.60 and 222.98% ROS generation in MCF-7 and MDA-MB-231 cells. The effects were confirmed by the reversal on N-acetylcysteine pretreatment. A higher intracellular ROS accumulation resulted in a loss of mitochondrial membrane potential (MMP). Replicating the study in vivo, the biodistribution and efficacy were monitored in tumor-bearing mice, screening of which was made via whole-body fluorescence. The persistence of visible fluorescence 48 h after delivery implied an elongated circulation for NCs. These effects were reciprocated in reduced 21-day TV for both MAN–PAA-conjugated and unconjugated DOX–MSNPs, supported by the attenuated fluorescence signal until 21 days, with a higher effect for DOX-loaded NCs. These observations suggested a sustained therapeutic efficacy of MAN–PAA–DOX–MSNPs, with null residual accumulation in the heart, liver, spleen, lungs and kidneys [189]. This study, thereby, offers multiple insights, the foremost being the MAN coating. Biocompatible entities similar to MAN are thereby highly suited to modulate the DOX therapeutic efficacy via MSNPs vis-à-vis stoichiometric equilibration.

Another recent endeavor using MSNPs for BC treatment involved their aptamer conjugation for DOX delivery to MDA-MB-231, SKBR3 and MCF-7 BC cells. The developed MSNPs were screened in neat, chitosan-coated (CTS–MSNPs) and DOX-conjugated chitosan (DOX–CTS–MSNPs) configurations, all exhibiting a spherical morphology with uniform surface textures. The TEM-ascertained PS were 68.2 ± 5.8 (MSNPs), 73.7 ± 6.5 (CTS-MSNPs) and 72.9 ± 3.6 nm (DOX–CTS–MSNPs), which increased to 84.6 ± 1.8, 152.2 ± 6.0 and 143.7 ± 2.5 nm, when screened as hydrodynamic radii via Photon Correlation Spectroscopy (PCS). The trends implied CTS’s hydrophilicity with no further rise in DOX loading. The reversal of anionic surface polarity for neat MSNPs (−43.6 ± 0.8 mV) to the cationic in CTS-coated (44 ± 4.3 mV) and subsequent DOX-loaded (38.8 ± 0.8 mV) states confirmed the CTS coating on MSNPs. Surface sensitivity probing via PCS revealed a 9.4% reduced *ζ*-potential for blank MSNPs on CTS carboxylation, which further increased by 4.67% on DOX loading. The need for CTS–MSNPs’ carboxylation was realized by a below-par response (28%) at 5.5 pH (amongst 5.5, 6.5 and 7.4) upon monitoring for 21 days. The carboxylation of surface CTS improved this extent to 80.21% (at 5.5 pH) for similar analysis duration. These distinctions were supported by 82.86% PV, 42.35% PS and 34.69% SA reductions in the CTS–MSNPs, suggesting an enhanced nanoscale effect, with 32.97, 60.82 and 79.97, 127.26% encapsulation and loading enhancements.

For the targeted DOX delivery, the EGFR, HER2 and random aptamers were conjugated to COOH-CTS NPs, all with a spherical morphology, uniform dispersion and a rough texture. The PS of aptamer-conjugated NPs were 80 ± 4.9 (EGFR), 80.6 ± 4.4 (HER2) and 80.3 ± 3.4 (random) nm, which increased to 208, 208.6 and 204 nm, on PCS screening. The surface charges of EGFR and HER2-aptamer conjugated NCs were −39.6 mV (each), while for random aptamers, these were 38.8 mV. The EGFR aptamer exhibited the maximum DOX internalization, as noticed via the sharp green fluorescence in MDA-MB-231 BC cells. Similar effects were noted for SKBR3 cells, corresponding to HER2 and EGFR conjugations. Therapeutic screening of DOX-loaded NCs revealed an IC_50_ of 15.36 ± 1.09 μM (the least) for the EGFR conjugation in MDA-MB-231 BC cells, with 26.98 ± 1.14 and 30.90 ± 1.16 μM extents for random aptamer and non-aptamer NCs. The HER2 and EGFR–aptamer-conjugated NCs exhibited a significant DOX therapeutic efficacy with 8.33 ± 1.14 and 6.33 ± 1.17 μM IC_50_ limits. The extents were greater for the random aptamer group (15.76 ± 1.31 μM) and unconjugated NCs (12.95 ± 1.34 μM). No such distinctions were noted for MCF-7 BC cells, having an IC_50_ of 7.79 ± 1.09 μM (HER2), 8.5 ± 1.12 μM (EGFR), 8.31 ± 1.10 μM (random aptamer) and 10.72 ± 1.10 μM (unconjugated DOX). Compatibility screening for NCs revealed maximum toxicity (24%) at 293.56 μg·mL^−1^ for the EGFR-conjugated carboxylated mode in MDA-MB-231 BC cells. Similar activities for HER2 and EGFR-conjugated NCs for SKBR3 cells ranged within 12–17%. The observations inferred a carboxylated CTS involvement for enhanced DOX uptake via bypassing the endo/lysosomal detection [190]. In all, this study illustrates the utility of EGFR and HER2–aptamer-conjugated COOH-CTS-MSNPs’ DOX delivery to MDA-MB-231 and SKBR3 cells. As TNBC contributes the most to BC mortality, the results offer significant promise for tackling BC severity.

Another elegant effort by Chang and colleagues used –COOH-functionalized and PEI, anisamide (AA) surface-engineered MSNPs for DOX delivery to MCF-7, MDA-MB-231 (human BC cells) and 4T1 murine mammary carcinoma cells in a pH-regulated manner. The DOX cargo (DOX-MSNPs-PEI-AA, DMPA) entry in the tumor cells was mediated via AA receptor endocytosis. The PEI dissociation induced DOX release from the MSNPs’ pores to the tumor cells’ cytoplasm, in the acidic environment of cellular lysosomes, due to PEI protonation. The MSN-COOH, MSN-PEI and MSN-PEI-AA NC configurations exhibited ~100.6, 117.7 and 146.9 nm PS, with an initial *ζ*-potential of −21.4 eV (–COOH functionality), changing to 20.4 eV (PEI-functionalized state) and a further reduction to −7.8 eV (AA-masked –NH_2_groups). Porosity inspection using Nitrogen Adsorption Isotherms revealed a reversible grade IV isotherm, with a ~380.747 m^2^·g^−1^, 1194 cm^3^·g^−1^ and 3.416 nm specific SA, PV and PD. These features corresponded to nanoscale attributes with FTIR-predicted structural intactness and TGA-inferred ~44% weight loss (volatile PEI groups). The DOX release from NCs was examined at pH 5.5, 6.8 and 7.4, revealing 32.29, 48.21 and 84.97% liberation, with minimal limits in normal tissues. Monitoring the biocompatibility, the MSNP–PEI–AA (no DOX) was evaluated separately in the chosen cell lines, wherein >80% viabilities were retained despite a 200 μg·mL^−1^ intake. The in vitro anticancer efficacy of DMPA was evaluated and revealed a dose-dependent viability suppression in response to free DOX and DMPA treatments. Targeted DOX delivery was inferred from the low viability of MDA-MB-231 than MCF-7 cells on exposure to similar DMPA extents for alike durations, indicating a higher expression of Sigma receptors (SRs) on MDA-MB-231 cells. This was complemented by a reduced fluorescence intensity (confocal microscopy) upon haloperidol (SR antagonist) administration to the 4T1 BC cells. Monitoring the anticancer effects in vivo (BALB/c female mice), the NC-assisted DOX was intravenously delivered to (i) the saline (control), (ii) free DOX, and (iii) DMPA groups. After 14 days of monitoring, the analysis of excised tumor tissues showed a significant DOX accumulation via DMPA-assisted delivery. To screen the DMPA carrier biocompatibility, the tumor weights of the treated animals were examined and noticed as consistently lower than for free DOX. These observations were complemented by insignificant pathological changes in the heart, liver, spleen, lungs and kidneys of the treated mice (H&E staining). This was further supported by <5% RBC hemolysis on 1 mg·mL^−1^ MSNP—PEI–AA intravenous injection. The efficacy of DMPA as a carrier was inferred by fluorescence distinctions, with a sharp fluorescence in the tumor and normal cells, in unaided DOX-administered mice. Contrary to this, for NC-delivered DOX, a high fluorescence was noted for tumor cells only, wherein overexpressed SRs facilitated DOX accumulation [191]. The recent conduct of this study projects immense suitability of MSNPs as drug carriers, as AA-targeting ligands and Sigma receptors on tumor cells have not been used in earlier attempts. It would be interesting to use such ligands with other CDs, as their overexpression on TNBCs could be a boost to improve the tumor cell internalization of drug cargos.

### 6.2. Combinatorial Drug Delivery Using Mesoporous Silica Nanoparticles

Recurrent resistance with CDs has reduced their clinical efficacy, thereby mandating a need for shifting to potential alternatives via predictive structure–activity relationships (SARs). In this regard, the combinatorial delivery of CDs (two or more) with sensitization agents (such as photothermal or X-ray irradiation) has lately gathered significant interest. The turning point aspect revolves around the long-known and practiced status of most of the CDs due to which their continued use inevitably receives a resistant response despite a stealth delivery via various NCs. To counter the competing therapeutic limitations, combining two or more CDs or the sensitizing agents with CDs has lately been an area of focus for developing potent drug delivery vehicles. This strategy benefits from multiple considerations, wherein the most important distinction is the reduced elimination on an in vivo scale. Newer combinations are not recognized as native, unlike the long-practiced CDs, whereby there stands a significant possibility of these being more effective. Secondly, for terminally ill or advanced stages (as in TNBC), rapid metastasis and fragility mandate accurate therapies. Using the combinatorial route, the concentration of the major CDs (often resistant) is effectively reduced, wherein the sensitization of healthy cells is substantially reduced. The combination of drugs no longer targets a single receptor on tumor cells and its unique chemical essence is the tunability with the mutated status of receptor proteins. Thereby, the chances of prompt elimination from physiological boundaries are much lower for a combinatorial regime. The most important benefit of combinatorial delivery is the possible use of bioactive and non-toxic nutraceuticals. Therefore, delivering a small proportion of CDs with these compounds could significantly improve their tumor cell internalization. Stronger therapeutic essence and dosage moderation are inevitable via NC-mediated delivery, as additional structural protection is accomplished. As for MSNPs, recent interest has superseded the success of Au and Ag NPs, both of which lack the porous morphology, and Ag NPs also exhibit high native toxicity that complicates their systemic elimination. Keeping these prospects in the background, the recent attempts (since 2017) using MSNPs for combinatorial drug delivery to treat BC are summarized in the following paragraphs. 

The single significant attempt from 2017 used TiO_2_-coated MSNPs as DOX carriers, with the assistance of 808 nm near-infrared (NIR) irradiation for thermal imaging-aided photothermal therapy (PTT) for BC chemotherapy. The study by Ren and colleagues highlighted the constraints of traditional white TiO_2_-like weak drug loading, inadequate UV light penetration in the tissues and the heating risks of 980 nm NIR to the healthy tissues. The preliminary screening of black TiO_2_ (B-TiO_2_) NPs revealed a PS of 25 nm (via TEM) with weak dispersion. Coating these NPs with mesoporous silica conferred a core–shell morphology to B-TiO_2_ NPs with a ~100 nm diameter. The prepared nanocomposites (B-TiO_2_ NPs in an MSNP core) were functionalized by the -NH_2_ group and subsequently reacted with-COOH in the FA, using an EDC–NHS conjugation. The recovered NCs were loaded with DOX, exhibiting a 150–260 nm PS and a <1 PDI for B-TiO_2_, NC, NC–NH_2_, NC–FA and NC–FA–DOX. The –NH_2_, FA and DOX conjugation to the NC surface was distinguished via the cationic surface sensitivity for B-TiO_2_ and NC–NH_2_comparedto anionic sensitivity for others. The DOX loading on NCs was screened as 5%, ~10-fold higher than uncoated B-TiO_2_. Screening the photothermal irradiation impact on DOX release from NCs, NIR irradiation at pH 5 enhanced the release from 60.6% (neat) to 91.3% in 48 h, conveying a pH and NIR-modulated liberation. The impact of FA targeting was distinguished in the fluorescence analysis, wherein NC–FA–DOX exhibited DOX deposition in the nucleus, while the non-targeted NCs (without FA) revealed little DOX in the cytoplasm, with inadequate nuclei localization. Screening the PTT influence on the therapeutic efficacy of DOX, a better performance for NC–FA was noticed than for NCs alone, with a 0–5 min NIR exposure on incubation with 75, 150 and 225 μg·mL^−1^ Ti, generating the 85.2 and 50.8% at 75 μg·mL^−1^, 76.5 and 10.9% at 150 μg·mL^−1^ and 34.1 and 5.5% at 225 μg·mL^−1^ viabilities for NC and NC–FA confirmations. 

Monitoring the influence of PTT on the anticancer effects of DOX, the analysis revealed cell death of 31.6% (by DOX alone) and merely 8% (by NC–DOX); and with NC–FA–DOX, this extent was 28.7%. On subjecting the treated cells to 5 min NIR irradiation, these limits increased to 44.8, 34.1 and 93.8%, implying the synergism of PTT and chemotherapeutic treatments. Screening the cell death mechanism in NC–FA and NC–FA–DOX-treated cells, a flow cytometry-driven inspection revealed most tumor cells being subjected to necrosis (death via photothermal-induced ablation). The in vivo screening of chemotherapy and PTT synergism was made in tumor-implanted xenograft mice, distinguished as (i) only DOX-treated, and (ii) treated with NC–FA, (iii) NC–FA + NIR irradiation, (iii) NC–FA–DOX and (v) NC–FA–DOX + NIR irradiation. An analysis of five mice, which were sacrificed to consolidate the PTT–chemotherapy synergism using HE staining revealed most cells to be dead in the DOX, NC–FA + NIR, and NC–FA–DOX groups, exhibiting nuclear or cell membrane damage. Greater tumor cell killing in the NC–FA–DOX + NIR group suggested a synergistic combination of PTT and chemotherapy rather than a singular effect. The observations were further strengthened by significant decrements in TV for the NC–FA–DOX + NIR group compared to the NC–FA + NIR and NC–FA–DOX groups. Moreover, no significant variations in body weight revealed the biocompatibility of the used NCs [192]. Thereby, this study established a synergism of TiO_2_ NPs’ toxicity with the FA-conjugated MSNPs for the localized therapeutic induction in the MCF-7 BC cells. Similar attempts could be made by co-delivering TiO_2_ NP-doped MSNPs with other CDs and some phytomolecules, which could further moderate the toxicity. Secondly, in place of Ti, the oxides of other elements (in the same group of the periodic table) could be a logical extension of this study. 

The year 2019 reported a sole major attempt relying on HER2-conjugated aptamer (HApt) cytotoxicity via cross-linking and subsequent HER2 translocation to cytoplasmic vesicles in the SKBR3 and MCF-7 BC cells. Relying on HER2 suppression-triggered impaired cell proliferation and impaired apoptosis, Shen and colleagues exploited the HApt targeting and antagonizing ability to augment the HER2-positive BC cell-specific toxicity of MSNPs. The carrier configuration involved HApt-functionalized pH-sensitive β-cyclodextrin (β-CD) capped with MSNPs conjugated with benzimidazole (MSN-BM), which was used to optimize a pH-driven DOX release. The purpose of including β-CD was to use its gatekeeper essence to deliver encapsulated DOX-HApt as the HER2-targeting biotherapeutics. The thorough physicochemical characterization of the optimized carrier revealed nanoscale attributes (via FT-IR, XRD, TEM and BET) with no structural arbitration. The DOX release was studied at pH 7.4, 6.4 and 4.5, with the maximum extent (>80%) being accomplished at pH 4.5, due to BM protonation-disrupted DOX–BM hydrophobic interactions. Comparing the NC-delivered DOX efficacy in HER2-positive SKBR3 and HER2-devoid MCF-7 cells, greater cytotoxicity, as well as enhanced uptake, was observed in the SKBR3 cells. The synergistic association was witnessed as the DOX-HApt co-delivery enabled higher toxicity in HER2-positive BC cells compared to either DOX or HApt. The reduction in cell viability for DOX-devoid NCs was15 ± 2%, for 100 μg·mL^−1^ NPs, while similar results for free DOX and aptamer-devoid NC–DOX (DOX extent: 3.6 μg·mL^−1^) were 8 ± 4% and 27 ± 6%. The MSN-BM/CD-HApt@DOX, however, reduced the viability by ~68 ± 6%, signifying a DOX and HApt synergistic essence [193].

The first major 2020 attempt focused on MSNP-facilitated combinatorial drug delivery to treat BC, using lactoferrin (Lf)-coupled NPs for simultaneous pemetrexed (PMT, a cytotoxic drug) and ellagic acid (a phytocompound) administration to MCF-7 human BC cells. The hydrophobic EA was physically encapsulated within the mesopores (via adsorption) assisted by electrostatic interactions between anionic EA and -NH_2_ group functionalized NCs. Contrary to this, the hydrophilic PMT was chemically conjugated to Lf coating via carbodiimide coupling, to prevent its early release and instantaneous toxicity aggravation. The dispersion analysis of PMT + EA-loaded NCs attributed their nanoscale distinction, with a PS of 284 nm (via DLS), 190–230 nm (via TEM imaging) anda 0.207–0.778 PDI, enabling a sequential drug release, first of EA, followed by sustained PMT liberation. The co-loading features were demonstrated well via differential scanning calorimetry (DSC) and powder XRD, showing a simultaneous drug loading that resulted in crystallinity loss. Structural coherence was inferred by FTIR spectroscopy with C=O ester of EA (1723 cm^−1^), C=O functionality of the second –COOH group of PMT (1690 cm^−1^), and the (1690–1630 cm^−1^) stretching frequencies of newly formed amide linkages. The synergistic association between PMT and EA was monitored via cytotoxicity studies, wherein 5:3 proportions of unaided EA and PMT exhibited 1.7 and 1.3-fold decrements in IC_50_ of EA (34 μg·mL^−1^) and PMT (26 μg·mL^−1^), complemented by a combination index (CI) of 0.975. These observations were further strengthened for NC-mediated EA delivery with a 23 and 20 μg·mL^−1^ IC_50_ for non-targeted and targeted NC configurations. Yet again, a CI value of 0.885 suggested EA and PMT synergism (less so than for unaided delivery). The role of Lf targeting in desired drug internalization in the MCF-7 tumor cells was screened via fluorescence intensity, which developed a faint green appearance, inhibiting the Lf receptors and incubating the cells with excess Lf before the uptake analysis. The results were supported by a flow cytometry inspection, wherein high MFI implied a higher intake of NCs by the tumor cells in the Lf-conjugated state [194]. Thereby, this study established Lf-targeted MSNPs as promising carriers for mediating synergistic PMT and EA anticancer actions in MCF-7 BC cells. The results indeed encourage evaluation in animal models and more aggressive BC subtypes.

A subsequent 2020 attempt towards using MSNPs for combinatorial therapies against BC involved DOX and a sonosensitizer, chlorin-e6 (Ce6), thereby combining sonodynamic therapy (SDT) with DOX–MSNPs. The prepared MSNPs (using CTAB as a surfactant and TEOS as a template) were optimized for DOX delivery via initial –NH_2_ and subsequent –COOH functionalization, which was observed in −31–20, 17–37 and −37 to −43 mV ranges o f*ζ*-potentials for native, –NH_2_ and –COOH-functionalized states, respectively. The variations suggested progressive -OH and –NH_2_ replacements, with a higher colloidal stability of –COOH-functionalized MSNPs. The PS of NCs using TEM was screened as 149.5 ± 12.2 nm, with the magnitudes reaching up to 300 nm in DLS measurements, due to conjugated water molecules. The DOX release was monitored without pH fixation, reaching an 80% extent after 10 h, due to the obstruction of the MSNPs’ pores. The release was slow but not sustained due to the NCs’ surface lacking mesopores. Uptake studies deciphered a therapeutic significance of MSN–DOX–Ce6 particles, as these were internalized in MDA-MB-231 cells to a higher extent (greater fluorescence) than DOX and Ce6 alone. The release mechanism was ascertained as “free-diffusion”, with the cells requiring co-culturing with drugs for 4 h before the sonication-energized antitumor efficacy of Ce6. Screening the antitumor efficacy of MSNP–DOX–Ce6, the carriers without DOX caused no change in cell viability, while MSNP–DOX–Ce6 with ultrasound (US), decreased it more than US–Ce6, DOX+Ce6+US or free DOX. The effects were reproduced in vivo (BALB/c nude mice) with substantial TW and TV decrements for the MSNP–DOX–Ce6–US group rather than Ce6–US or DOX alone [195]. Thereby, this study established an MSNP–DOX–Ce6 and US synergism for DOX efficacy in MDA-MB-231 TNBC treatment.

In perhaps the last major attempt from 2020, Moodley and Singh used MSNPs neat, in a CTS-functionalized state and with (2–5)% PEG functionalization for DOX and 5-FU co-delivery to MCF-7 BC cells. The as-synthesized MSNPs (CTAB as a template, TEOS as a silica precursor) were functionalized with CTS and PEG. The optimum functionalization was attained by the reversal of the anionic surface sensitivity (for MSNPs) to the cationic regime in the CTS and PEG-functionalized state. The TEM-ascertained PS for MSNPs ranged between 37 and 66 nm, while the 12–215 nm hydrodynamic diameters were due to the enhanced aqueous interactions of the silanol groups. The 710.36 m^2^·g^−1^ SA and 1.74 cm^3^·g^−1^ PV implied a nanoscale sensitivity of the prepared NPs, prevailing as graduated mesoporous spheres for drug loading and release. Encapsulation of 5-FU and DOX into the PEG-CTS@MSNPs determined a combined drug loading of 20–30%, due to their differential interactions with the carrier constituents. With a higher hydrophobicity, DOX interacted with hydrophilic PEG and was internalized to a greater extent than 5-FU, which is attracted to CTS and –NH_2_ groups and is internalized afterwards. The release kinetics were monitored at pH 7.4 and 4.2, with the former resulting in a higher 5-FU + DOX liberation of 21% and 52% for NCs functionalized with 2% PEG. At pH 4.2, the release extents were 29 and 59% for 2 and 5% PEG-functionalized NCs and for 5-FU, while for DOX, these were 62 and 53%. The distinctive release extents of 5-FU and DOX were attributed to the pH-dependent 5-FU protonation, wherein a 4.2 pH caused Fickian diffusion for both drugs from 2% PEG-functionalized NCs with negligible erosion. For 5% PEG-functionalized NCs, the diffusion and erosion constants played a complementary role to modulate DOX release. Slower drug diffusion was reasoned by investigators due to DOX interaction with the MSNPs matrix and aqueous medium, besides the pH and PEG coating on the MSNPs. Screening the antitumor efficacy, the MCF-7 BC cells treated with 5-FU + DOX revealed maximum apoptosis with 2% PEG-functionalized NCs, causing DNA condensation and fragmentation. A cell cycle analysis revealed enhanced populations in S and G2/M phases, inducing DNA damage and cell cycle arrest, correlating with most of the cells undergoing apoptosis and fragmentation. These were witnessed in AO/EB-stained cells, hinting at the toxicity emanated by 5-FU and DOX-co-loaded MSNPs [196]. 

#### 6.2.1. Research Attempts from 2021

Moving on to 2021, the first major study involved BC treatment via the co-delivery of DOX-loaded, redox-sensitive MSNPs and DNA aptamers (AS1411). The optimized carrier configuration used AS1411 (aptamer) and a small interfering RNA (siTE2) as a gatekeeper on MSNPs via redox-sensitive disulfide bonds. To make the functional NCs, the DOX-loaded MSNPs-SH (via disulfide linkers) were capped with siTIE2 and AS1411 to retrieve MSNPs-siRNA/Apt. The optimum NC configuration was screened via TEM, revealing a spherical shape, 90–110 nm average diameter and nucleic acid shells enclosing MSNP cores. The average pore sizes of neat MSNPs, disulfide-conjugated MSNPs (MSNPs-SH) and si-RNA–aptamer-conjugated MSNPs (MSNPs-siRNA/Apt) were 2.9, 2.4 and 1.7 nm, suggesting the nanoscale attributes. Successful siRNA and DNA aptamer conjugation was determined via *ζ*-potential decrements for native MSNPs. The DOX loading on NCs was ascertained using fluorescence spectroscopy, revealing an extent of ~8.5%. The 20:1:1 (*w*/*w*) MSNP, SH-siTIE2 and SH-AS1411 stoichiometries enabled an optimum nucleic acid conjugation, with a 90.7% efficacy. The release kinetics of DOX from the MSNP–siRNA–Apt–DOX revealed maxima (~80%) with 10 mM GSH and nucleolin, typical traits of a cellular microenvironment. 

The release of DOX was ascertained as GSH concentration-dependent, while similar inspections for siRNA implied its burst release until 5 mM GSH, conveying suitability of >5 mM GSH for efficient siTIE2 liberation via disulfide bond cleavage. Monitoring the uptake studies of DOX-loaded NCs using confocal microscopy and fluorescence spectroscopy in nucleon-positive (MDA-MB-231 cells) and negative (HEK293T cells), the NCs devoid of AS1411 functionalization lacked the tumor cell-penetrating ability. Contrary to this, the MSNP-siTIE2/Apt-DOX internalized the MDA-MB-231 BC cells to a much higher extent, complemented with a high MSN-siTIE2/Apt-DOX fluorescence. The pharmacokinetics of NC-delivered DOX also exhibited a significant improvement with comparatively high fluorescence intensity from primary tumor sites for MSNP–siTIE2–Apt–DOX compared to free DOX and siTIE2 lacking NCs. The MSNP–siTIE2–Apt–DOX caused a 77.4% IC_50_ decrement compared to unaided DOX; the potency was highest amongst all NC configurations. The efficacy of NC-delivered DOX was also monitored in tumor-bearing mouse xenograft models, whereby no significant changes in major organs and tumors (H&E staining) were noticed. Only MSNP–siTIE2–Apt–DOX exhibited large necrotic regions within tumor cells. The effects were also pronounced vis-à-vis metastasis inhibiting the ability of DOX-loaded NCs, screened using bioluminescence imaging. The decrement in MSNP–DOX–siTIE2 and DOX–siTIE2 bioluminescence indicated reduced metastasis in mice models. Metastatic nodules were screened and MSNP–siRNA–Apt–DOX-administered mice exhibited the least metastasis, with fewer metastatic foci in the lungs compared to free DOX and DOX–siRNA configurations [197]. This study thereby demonstrated the improved therapeutic potency of DOX via the combinatorial delivery with siRNA, an antisense molecule vulnerable to oxidative damage. Using MSNPs as carriers to co-deliver siTIE2 protected them from enzymatic damage and prolonged their therapeutic essence for a synergistic response, moderating DOX side effects and troublesome patient sensitization. 

The last major 2021 effort aimed at co-delivery characterized MSNP-mediated BC treatment focused on the sequential and concurrent administration of niclosamide (NIC, a Wnt signaling inhibitor) with DOX (a CD) in the MDA-MB-231 (TNBC), SKBR3 (HER2-positive) and MCF-7 (hormone receptor-positive) BC cells. The as-prepared MSNPs (via reverse microemulsion) were surface-conjugated with CTS using a silane cross-linker to confer pH sensitivity before DOX loading, finally providing DOX–CTS–MSNPs. To confer a pH-sensitive drug release ability, the –NH_2_ groups in CTS were converted to –COOH, enabling COOH–CTS–MSNPs. The morphology inspection of these NPs revealed a spherical shape with a homogeneous dispersion. The PS of all modes was <100 nm, with a cationic surface in CTS-coated NPs being changed to an anionic regime for COOH functionalization, as well as with NIC and DOX conjugation. The unchanged surface sensitivity from COOH conjugation to the drug-loaded NCs implied a drug loading within the pores. The flow cytometry-ascertained uptake for COOH–CTS–MSNPs revealed a bright green emission in all cells, with a time-dependent increment (at 3 and 6 h). Screening the antitumor activity in MDA-MB-231 cells, the NIC-loaded MSNPs developed 10.25 and 8.01 μM as IC_50_, upon 24 and 48 h monitoring, while similar extents for DOX were 14.15 and 1.44 μM. A lower IC_50_ with DOX at 48 h revealed its target-site-specific modulated activity, via NC-driven stealth delivery.

A drug-lacking carrier (COOH–CTS–MSNPs) was also examined for toxicity, but no significant population decrements were noticed at24 and 48 h. The NIC + DOX actions via NCs were screened in sequential and concurrent modes, both demonstrating concentration-dependent overall cytotoxicity. In sequential mode, 6, 8 and 10 μM NIC were combined with 8–12, 6–12 and 4–12 μM DOX, all causing >50% cell death. Similar observations were made for 2.5, 5 μM NIC with 1–4 μM DOX, 10 μM NIC with 0.5–4 μM DOX and 7.5, 15 and 20 μM NIC with 0.25–4 μM DOX. In the combinatorial mode, 1, 4 and 6 μM NIC with 8 μM DOX (CI < 1) and 8 and 10 μM NIC with 6, 8, 10 and 12 μM DOX were shown to be synergistic. In the concurrent regime, 24 stoichiometries were determined to be synergistic, including 2.5 and 10 μM NIC, each with 4 μM DOX, 5 μM NIC with 1, 1.5 and 4 μM DOX, 7.5 μM NIC with 0.25 and 1–4 μM DOX, 15 μM NIC with 0.5–4 μM DOX and 20 μM NIC with 0.25–4 μM DOX. These observations suggested a NIC and DOX synergism for TNBC therapy, with a better response in concurrent combinations. In HER2-positive SKBR3 cells, too, the NIC + DOX-loaded COOH–CTS–MSNPs exhibited a dose-dependent cytotoxicity at 24 and 48 h. 

The IC_50_ extents for NIC-loaded NCs were 31.5 and 13.53 μM at 24 and 48 h, while those for DOX were 3.41 and 1.53 μM. No significant toxicity of blank NC was observed for any of the cells. In combinatorial mode, the entire 36 sequential and 42 concurrent combinations resulted in >50% cell death, with the former developing a lower CI. Surprisingly, low concentrations of NIC and DOX exhibited a stronger synergism, arguing well for the reduced risk of high CD dosages. Finally, an analysis in HER2-positive MCF-7 BC cells revealed IC_50_extents of 35.6 and 5.2 μM for NIC and DOX (separately)-loaded NCs. Blank NCs did not exhibit any toxicity to MCF-7 cells but, of the screened 36 sequential combinations, 15 resulted in > 50% cell death. These included 15 μM NIC with 6 μM DOX, 20 μM NIC with 5 and 6 μM DOX and 25 and 30 μM NIC (each) with 1–6 μM DOX. Only nine of these combinations, carrying 1, 2, 5 and 6 μM DOX with (each) 25 and 30 μM NIC–NPs and 20 μM NIC with 6 μM DOX exhibited synergism. No cytotoxicity was observed in the MCF10A cells, in response to COOH–CTS–MSNPs, demonstrating their biocompatibility. The results, thereby, provide multiple optimized stoichiometries for NIC + DOX for the BC subtypes, making it prudent to screen these combinations in animal models and at least in terminally ill patients. Furthermore, as the NIC and DOX synergized in multiple stoichiometries, screening the compounds structurally similar to NIC could advance the therapeutic potency [198]. This could even further moderate the significant DOX extent for an efficient BC treatment. 

#### 6.2.2. Select Research Attempts from 2022

Moving on to 2022, the first attempt for BC treatment via combinatorial drug delivery using MSNPs involved gallium (III) nitrate (Ga(NO_3_)_3_) and CURC (as a complex) to MCF-7 human BC cells. A collaborative outcome of India and Poland, the study commenced with the design of a passive targeting optimized NC wrapped around the HMSNPs with 3-aminopropyltriethylsilane (APTES). The APTES-coated HMSNPs were loaded with Ga(NO_3_)_3_ and CURC separately, as well as their mutual complex (GNCC). The drug-loaded HMSNPs were screened via nitrogen sorption, revealing an altered hysteresis loop. The 37 and 56% decrements in specific SA, pore diameter and PV from HMSNPs’ (neat) state to GaC-loaded HMSNPs and Ga(NO_3_)_3_ and CURC co-loaded NCs revealed enhanced nanoscale functionality. Loading of Ga(NO_3_)_3_ and CURC was also inferred by the varied surface sensitivity, which changed from cationic (neat HMSNPs) to anionic (both for Ga(NO_3_)_3_ and GaC-loaded APTES-coated HMSNPs). Similarly, structural coherence and optimum binding were predicted by XRD, FTIR spectroscopy and X-ray photoelectron spectroscopy (XPS). The drug release ability of Ga-HMSNPs and GaC-HMSNPs (both APTES-coated) were monitored at pH 3, 6 and 7.4, wherein GaC exhibited a release of 87.5% (at pH 7.4), 69.57% (at pH 6) and 59.24% (at pH 3) at 72 h, while similar extents for Ga(NO_3_)_3_ alone were 81.3, 71.15 and 60.25%. A striking contradiction is the drug release from GaC at a pH of 7.4, unlike the <7 pH discussed in most studies. This distinction demonstrates the structural uniqueness of CURC–Ga interactions, creating an interest in a possible Ga replacement with other group members in the periodic table. Encouragement in this direction already prevails, with PBA as a ligand for CUM delivery through MSNPs. The mechanisms for GaC actions were screened via molecular docking, whereby calnexin, HSP60, protein disulfide isomerase, phosphoinositide-dependent protein kinase (PDK), caspase 9, Akt1 and PTEN were noted as significantly altered. 

The in vitro antitumor inspection of Ga, as well as GaC-loaded HMSNPs revealed a dose and time-dependent response, with GaC-loaded NCs exhibiting a 25 μM IC_50_ against the 40 μM with Ga alone. The mechanism of Ga and GaC-initiated apoptosis, via western blotting, revealed enhanced caspases 9,6, cleaved caspase 6, PARP, poly-ADP ribose polymerase and GSK 3β, glycogen synthase kinase-3. The reduced functioning of mitochondrial proteins (prohibitin1, HSp60 and SOD1) with the phosphorylation of oncogenic proteins (Akt, c-Raf, PDK1) together resulted in the elimination of tumor cells [199]. Thereby, this study established that aGa(NO_3_)_3_ and CURC combination could enhance the apoptosis of MCF-7 human BC cells in a mutually supporting manner. Combinations with CURC could be prioritized in multiple stoichiometries, as curc is edible and hardly poses any toxicity to a reasonable extent. Added incentives of using CURC and similar biocompatible nutraceuticals is their low toxicity, wherein co-delivering their major proportion with minor quantities of CDs modulates the tumor cell internalization via reduced non-specific spillage, noticed frequently in unaided CD delivery. Our earlier studies present much evidence for this claim, wherein CURC dispersed in oil–water micro/nanoemulsions have enhanced the free radical scavenging activity of 5-aminoindazole-functionalized graphene oxide (GO) and enabled pro-oxidant-to-antioxidant activity of GO alone [200,201]. Similarly, a combination of CURC nanoformulations with polyphenol-capped Ag NPs has considerably improved their free radical scavenging on equipartitioning the philicphobic force gradients [202]. The long-term usage of CURC with well-understood structure–function correlations further augments its inclusion as a therapeutic ingredient, establishing its co-delivery with CDs in varying stoichiometries as significantly better [203,204]. 

In a more recent and worthwhile effort, Predarska and colleagues used >1000 m^2^·g^−1^ specific SA, 4–30 nm PS, thick framework and MSNPs exhibiting a small crystallite size (SBA-15 silica) for loading platinum conjugates over cisplatin (DDP) scaffolds with caffeic and ferulic acid derivatization at their axial locations. The therapeutic efficacy of DDP conjugates via MSNPs was screened in MDA-MB-468 and HCC1937 TNBC cell lines, BT-474 and MCF-7 HER2 positive BC cells and on in vivo xenograft 4T1 tumors in BALB/c mice. The prepared MSNPs carried TEOS as a silica precursor and P123 as a template and exhibited a 200–800 nm range in size. Efficient loading of platinum derivatives was screened by N_2_ physisorption, wherein a type IV isotherm with a hysteresis loop suited for capillary condensation and ordered mesoporous morphology was suggested. The optimized NCs were named SBA-15-1 (for DDP–acetyl-guarded caffeate), SBA-15-2 (DDP–acetyl-guarded ferulate) and SBA-15-3 (DDP–ferulate) (Figure 15). The drug release from DDP derivatives was monitored at pH = 7.4, liberating 18 and 15% contents for SBA-15-1 and SBA-15-2 at 72 h whereas SBA-15-3 exhibited a >70% release extent. The slow release from NCs (with SBA-15-3) argued well for a moderation of instantaneous DDP toxicity. The trends were complemented by IC_50_ distinctions, with a mere 0.1 nM extent for SBA-15-1 against the BT-474 cells. The underlying mechanisms implied a significant apoptotic induction as the cause of reduced cell viability, alongwith a substantial accumulation of fragmented DNA in sub-G cell cycle phase and sharp caspase activation. The application of AO assay revealed an autophagic induction with a further reduction in cell viability on autophagy suppression, via subjection to 3-methyladenine (3MA), conveying an autophagy-conferred cell defense from destructive stimuli. The in vivo screening on the 4T1 orthotopic model (in syngeneic BALB/c mice) revealed a ~50% TV reduction for SBA-15-1 treatment (at 35 mg·kg^−1^), which was only 20% for 10 mg·kg^−1^ acetyl-guarded caffeate. The relative extents of necrosis and mitosis were reduced in all tumor cells with 14 and 12% necrosis for SBA-15-1 and DDP. No hepatotoxicity, nephrotoxicity or inflammation was observed in the treated animals (unlike unaided DDP therapy), except the dilatation of minor venules in SBA-15-1-administered mice [205]. The study, therefore, offers significant insights, with DDP being the frontline drug; a moderation of its dosage could be important for its future reliability. 

A few more studies are listed in Table 2, wherein interest in the combinatorial mode determines the structural compatibility of MSNPs. The latest of these was reported in November 2022, wherein DOX-loaded MSNPs were combined with PTX micelles, via Schiff-base-modulated pH fixation. Though all aspects could not be traced, this attempt used arginine-glycine-aspartic acid (RGD) peptide as a targeting agent, enabling 81% PTX and 65% DOX release upon 72 h monitoring. Yet again, this study also comprised a rigorous physicochemical inspection using DLS, UV–Vis, FTIR and NMR spectroscopies, BET and FE-SEM. 

## 7. Future Challenges in the Therapeutic Usage of Mesoporous Silica Nanoparticles

Despite immense progress and success for improved drug delivery in various tumors, some concerns of MSNPs still hinder their use as drug delivery vehicles. The first and foremost aspect of concern pertains to the aqueous hydrolysis-driven degradation of silica and silicon-based materials. It is thereby quite logical to ensure a minimum contact with residual water or humid air, which must be accounted for in each step. Therefore, it is pertinent to screen via adequate characterization that the structure of the host material (during the preparation method of carrier) is retained in an unaltered form in case water prevails. The degradation of the carrier material could impair the gate-based drug release from MSNPs, besides complicating the drug release in the in vivo conditions [211]. The second challenge in the MSNPs’ use for drug delivery relates to their biocompatibility and toxicity in course of the drug delivery to cardiovascular disorders. This constraint has been lately addressed by coating the drug-loaded MSNPs with the cell membrane biomimetic NPs, which not only evade the immunological clearance but also improve their targeting specificity. As the outer periphery of functional drug-loaded silica NCs resembles cellular membranes, these are recognized as self-assembled, owing to which, their toxic side effects are minimized. Serious concerns of this sort have been reported for atherosclerosis treatment [212]. Rigorous characterization and a vigilant precursor–template stoichiometry selection via in silico computational tools could minimize these limitations. The optimization of binding energy and relative stabilities via computational methods such as Density Functional Theory and Docking modules could emerge to be decisive in this respect. A cautionary reference to databases of varied configurations and their likely outcomes in the in vivo inspection could improve the biocompatibility of MSNPs in different environments. This could be accomplished by the judicious selection of precursors and templates and a stability monitoring of functionalized states vis-à-vis timing and extent of drug loading.

## 8. Summary and Takeaway Message

We have comprehensively discussed the manifold nanoscale functionalization-driven structural tolerability aspects of MSNPs as drug carriers. The manifested provisions of varying pore sizes, PV and specific SA encompass the tuning mechanisms of MSNPs as drug vehicles. For BC treatment, targeted delivery exhibits a greater success using MSNPs as delivery vehicles, attributed to the enhanced internalization in the tumor cells. The judicious optimization of stoichiometry, CDs and phytocompounds’ combinatorial activities with the robustness of non-covalent-mode ligand functionalization manifests successful targeted drug delivery via MSNPs. The rigorous assessment of recent studies on MSNP-mediated drug delivery to BC cells distinguishes DOX as the most used CD, with more attempts in the singular mode. Figure 16 describes the DOX and some other CDs’ potent structural distinctions, wherein TAM, DDP and 5-FU have been significantly used in singular modes. The high toxicity of DDP and TAM is caused by their aggressive hydrophobicity, making them poorly bioavailable (Figure 16d,e). Delivering these through NCs has been attempted and extending this for their combinations with phytocompounds could reduce the treatment durations and intake drug extents. The incentive of phytoactive compounds is their non-toxicity-driven enhanced tumor internalization, which could moderate the CDs’ dosages and reduce their toxic residual effects (on normal tissues). 

Many combinations remain unattempted to date, whereby better outcomes for designing effective drug regimens could be accomplished. However, the timely detection of tumors is a prerequisite, irrespective of the treatment accuracy. Although theranostic mechanisms using MSNPs are in search oftimid BC detection, the distinctions between in vivo and in vitro conditions necessitate betterment in this domain. Finally, the significantly encouraging findings of recent studies (cell lines and animal models) raise hopes of their perpetuation in clinical trials and terminally ill patients. Accuracy towards studying SARs and reproducibility of in vitro studies with a critical note of previous attempts collectively hold the key to advancing the therapeutic tunability of MSNPs as self-assembly-modulated drug carriers. 

## Figures and Tables

**Figure 1 pharmaceutics-15-00227-f001:**
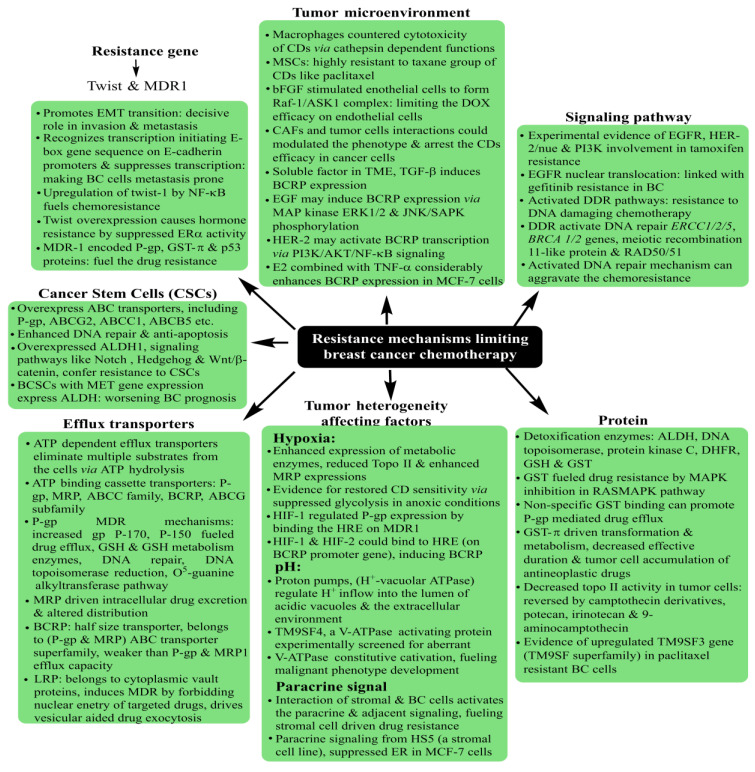
Summary of breast cancer chemoresistance mechanisms, collectively suggesting a need for combinatorial therapies and nanocarrier-assisted delivery as the counter alternatives [Abbreviations: BC: breast cancer, BCRP: breast cancer resistance protein, MDR: multiple drug resistance, CDs: chemotherapeutic drugs, HIF-1: hypoxia inducible factor, GST: glutathione S-transferase, BCSCs: breast cancer stem cells, HRE: hormone receptor element, P-gp: P-glycoprotein, *ERCC*-1: DNA excision repair *protein* MAPK: mitogen-activated protein kinase, GSH: glutathione, RASMAPK: Ras/mitogen-activated protein kinase, DDR: DNA damage response, ER: estrogen receptor, DOX: doxorubicin, E2: estradiol, MET gene: mesenchymal–epithelial transition factor, ABCC: ATP-binding cassette C, ABCG: ATP-binding cassette, subfamily, ALDH: aldehyde dehydrogenase, Topo II: topoisomerase II, DHFR: dihydrofolate reductase, RAD50/51: separate pathways that collaborate to allow cells to survive in the absence of telomerase, TM9SF3: transmembrane 9 superfamily member 3, JNK/SAPK: c-Jun N-terminal kinases/stress-activated protein kinases, MRP: multidrug resistance proteins, MET: mesenchymal–epithelial transition, EMT: epithelial to mesenchymal transition, EGFR: epidermal growth factor receptor, HER2: human epidermal growth factor receptor-2, LRP: lung receptor-related protein].

**Figure 2 pharmaceutics-15-00227-f002:**
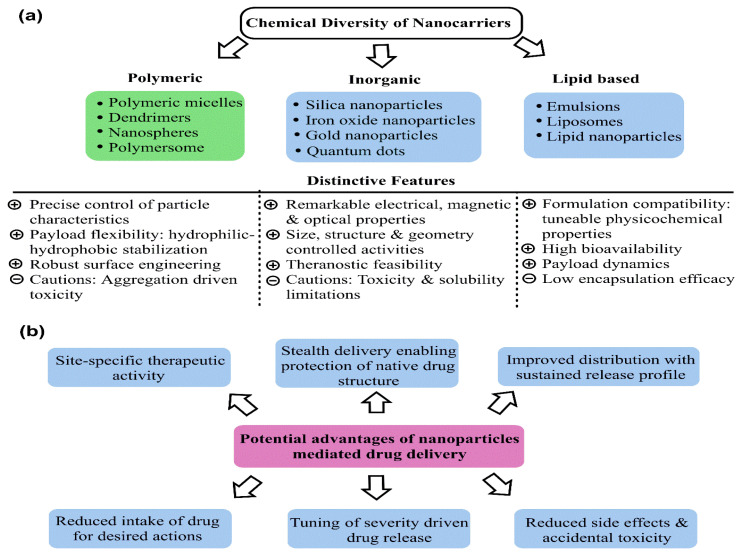
(**a**) Distinctive prospects of the various nanoparticles used as drug delivery carriers, and (**b**) Exclusive advantages of using nanoparticles as drug vehicles, reducing the risk of chemoresistance.

**Figure 3 pharmaceutics-15-00227-f003:**
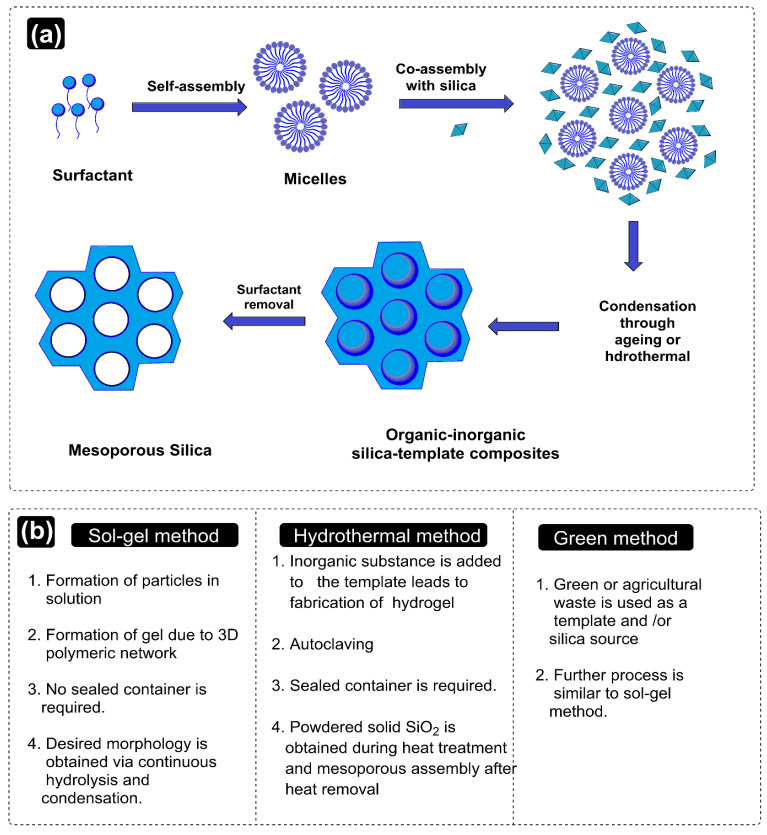
(**a**) Representative stages for surfactant–template-mediated fabrications of mesoporous silica nanoparticles. (**b**) Distinguishing features of sol–gel, hydrothermal and green methods.

**Figure 4 pharmaceutics-15-00227-f004:**
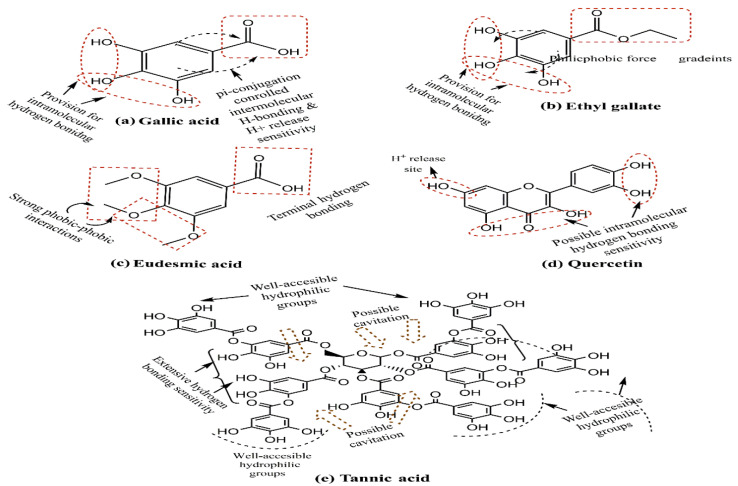
Structure-function dynamics of select templates for controlling particle sizes of mesoporous silica nanoparticles. It is usually that most of the used molecules exhibit plentiful hydrogen bonding interaction possibilities with ubiquitous –OH groups in their structure.

**Figure 5 pharmaceutics-15-00227-f005:**
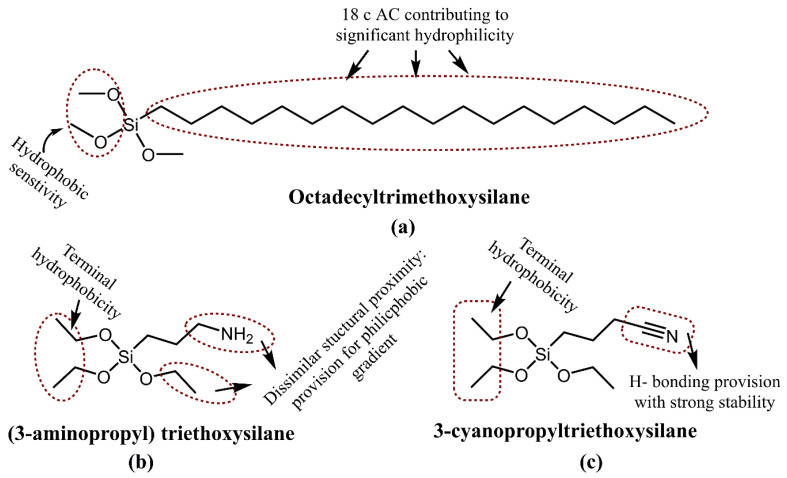
Structural distinctions of octadecyltrimethoxysilane (OTMS), (3-aminopropyl) triethoxysilane (APTES) and 3-cyanopropyltriethoxysilane (CPTES). Unlike APTES and CPTES, the unblocked hydrophobicity in OTMS confers a greater monodispersing ability driven by steric stabilization. The APTES and CPTEs are both only partially hydrophilic, with terminal –NH_2_ and N substitutions.

**Figure 6 pharmaceutics-15-00227-f006:**
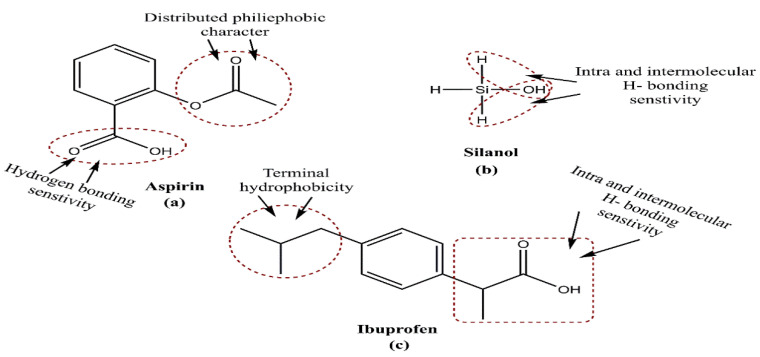
Chemical structures of aspirin, silanol and ibuprofen; the –NH_2_ functionalization of silanol could bind aspirin and ibuprofen more strongly than non-functionalized. This could be a reason for their steady release when delivered with –NH_2_-functionalized silanols.

**Figure 7 pharmaceutics-15-00227-f007:**
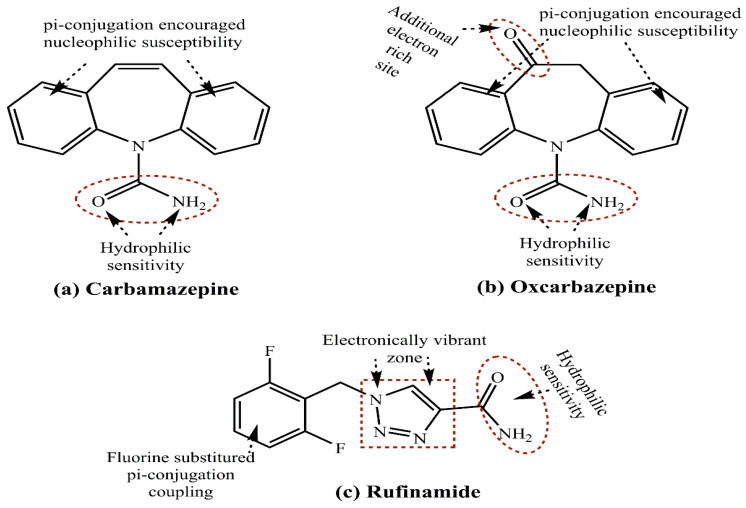
Structural distinctions of (**a**) carbazepine, (**b**) oxacarbazepine and (**c**) rufinamide. The presence of amide functionality in all these drugs provides a structure–activity-based understanding of their antiepileptic potency.

**Figure 8 pharmaceutics-15-00227-f008:**
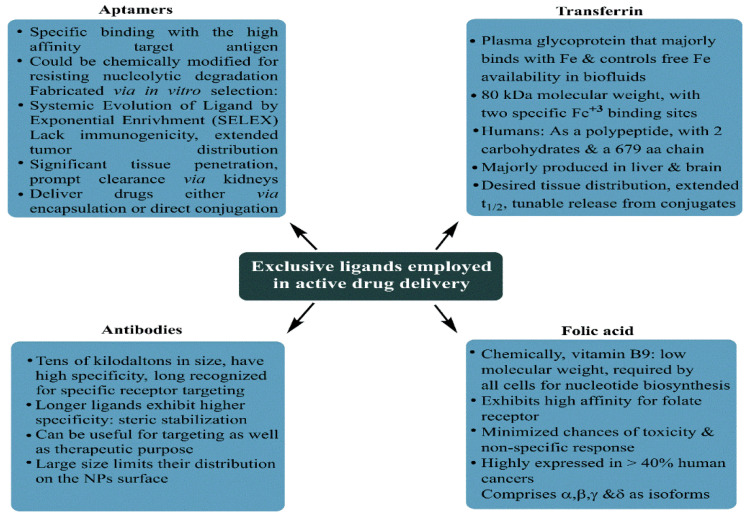
Potential ligands withan anticipated suitability for active drug delivery with mesoporous silica nanoparticles. While antibodies have a long history ofusage, recent suitability in folate receptors alongwith the architecture of aptamers has minimized toxicity and non-specific responses.

**Figure 9 pharmaceutics-15-00227-f009:**
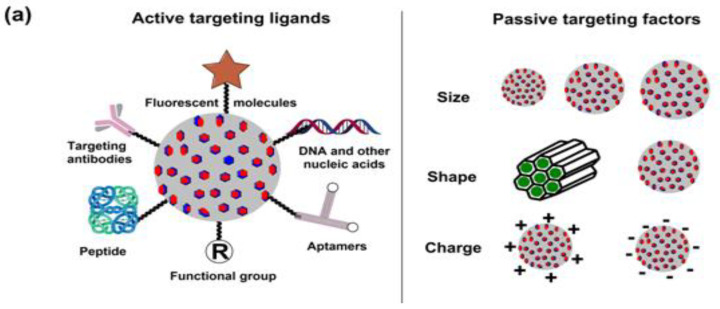
(**a**) Pictorial distinctions of active and passive drug delivery mechanisms in nanoparticles, with the former working through ligand specificity, while the latter relies more on physicochemical properties, such as surface charge, size and morphology. (**b**) Working mechanisms of active and passive drug delivery with the latter being aimed at gathering adequate drug amounts in the tumor cell vicinity, which are subsequently internalized within the cancer cells via physicochemical drifting.

**Figure 10 pharmaceutics-15-00227-f010:**
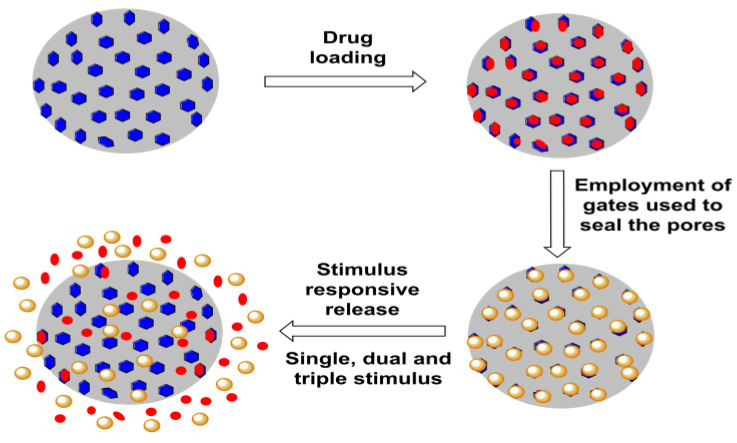
Schematic representation of pore-size-driven drug loading and the stimulus-modulated drug release mechanism from MSNPs. The stimulus-aided release mechanism has proved excellent in improving the efficacy of chemotherapeutic drugs via moderated dosages.

**Figure 11 pharmaceutics-15-00227-f011:**
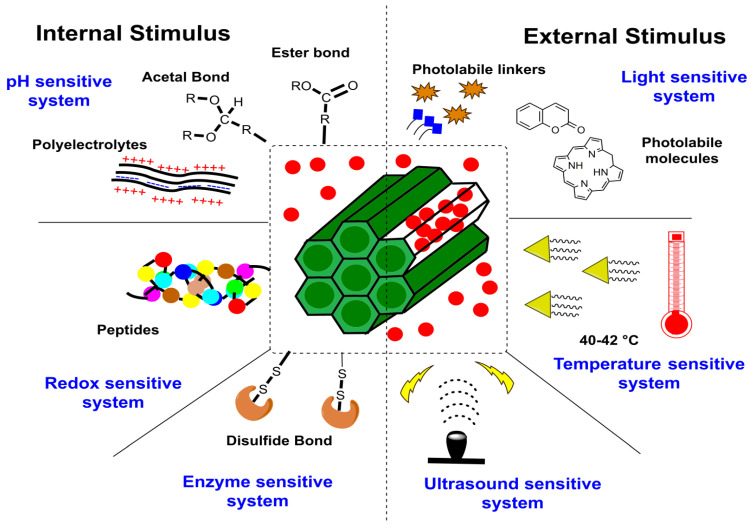
Different stimuli-based gatekeepers that keep the drug guarded inside the mesoporous silica nanoparticles and trigger an on-demand release.

**Figure 12 pharmaceutics-15-00227-f012:**
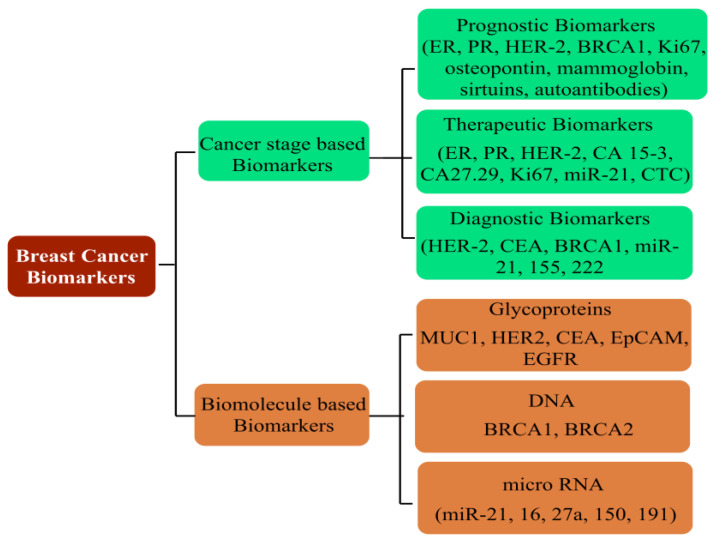
A summary of potential breast cancer markers diversified via (i) cancer-cell-stage-driven overexpressed surface markers and (ii) those which are expressed in conjugation with biomolecules. The former is rather more specific (exclusive to BC), while biomolecule-based markers are more reliable as therapeutic targets.

**Figure 13 pharmaceutics-15-00227-f013:**
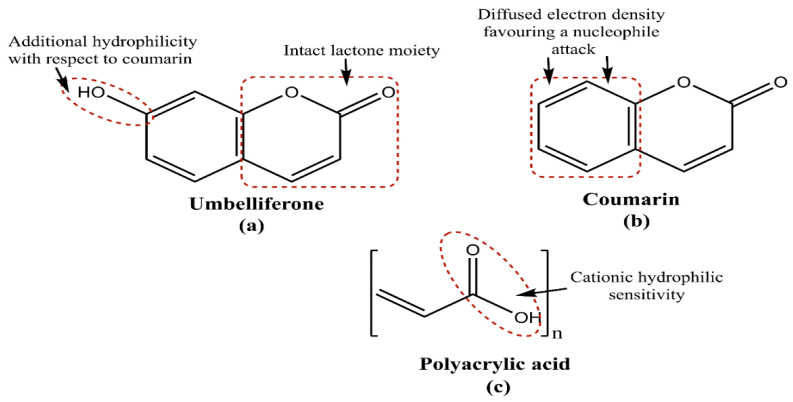
Chemical structures of (**a**) umbelliferone, (**b**) coumarin and (**c**) polyacrylic acid, depicting their philic–phobic sensitivities for an interaction likelihood.

**Figure 14 pharmaceutics-15-00227-f014:**
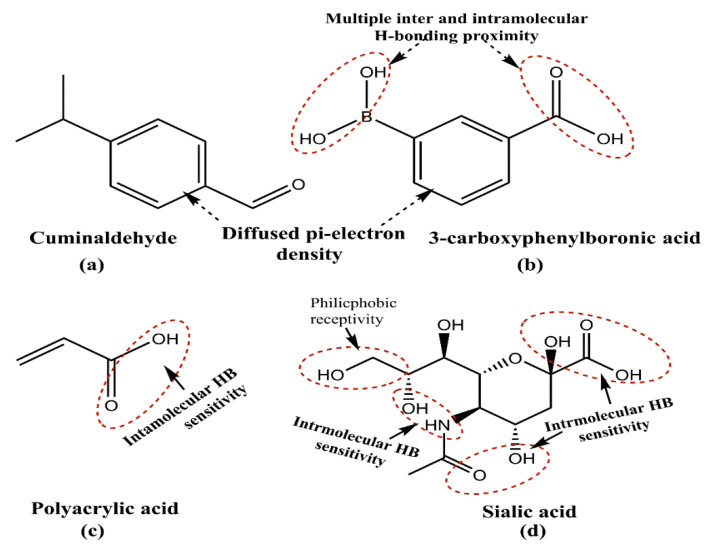
Chemical structures of (**a**) cuminaldehyde, (**b**) 3-carboxyphenylboronic acid (PBA), (**c**) polyacrylic acid (PAA) and (**d**) sialic acid. While cuminaldehyde and PBA are endowed with pi-electron densities favoring an electrophile attack, the pulsating hydrogen bonding susceptibilities in PBA, PAA and sialic acid receptors imply that non-covalent interactions provide a higher suitability for sustained drug release through nanocarriers.

**Figure 15 pharmaceutics-15-00227-f015:**
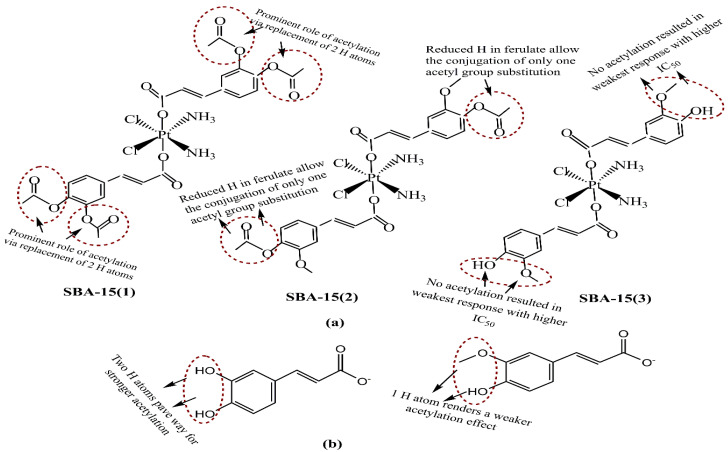
(**a**) Structural distinctions of acetylated caffeate and ferulate, with SBA-15(1) exhibiting the highest impact for acetylation and, hence, the anticancer efficacy, as opposed to SBA-15(3), for which the maximum IC_50_ were noticed. (**b**) Chemical structures of caffeate and ferulate.

**Figure 16 pharmaceutics-15-00227-f016:**
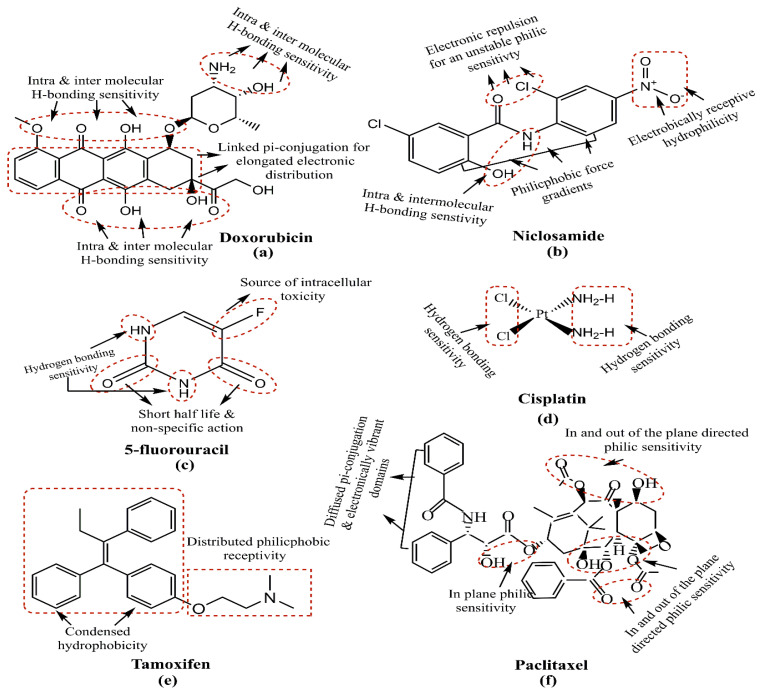
Structural distinctions of (**a**) doxorubicin, (**b**) niclosamide, (**c**) 5-fluorouracil, (**d**) cisplatin, (**e**) tamoxifen and (**f**) paclitaxel. The high toxicity of doxorubicin, cisplatin, 5-fluorouracil and paclitaxel concur with a need for their nanocarrier-mediated delivery. Alternatively, co-delivery (without nanocarriers) with natural phenols such as curcumin, hesperetin, quercetin and others (all much less toxic) could moderate the dosages of listed chemotherapeutic drugs.

**Table 1 pharmaceutics-15-00227-t001:** Prominent surface receptor proteins overexpressed on breast cancer cells [167,168].

Name of Receptor	Physiological Distinctions	Potential in Breast Cancer Treatment
ErbB tyrosine kinase receptors	Widely investigated growth factor receptors in BC that consist of four homolog receptors: ErbB1 (HER1/EGFR), ErbB2 (HER2/neu), ErbB3 (HER3) and ErbB4 (HER4). EGFR and HER2 are overexpressed in 15–20% and 20–25% of BC cells. The BC cells overexpressing ErbB receptors demonstrate aggressive clinical behavior.	Enhanced cytotoxicity of CURC delivered via EGFR targeting GE11 peptide-conjugated CURC-loaded PLGA-PEG NPs (in MCF-7 cells). HER2 Ab-conjugated PTX and rapamycin-loaded glycerol monooleate-coated magnetic NPs exhibited enhanced uptake in MCF-BC cells with 24-fold reduced toxicity and 3-fold lower IC_50_ extent (for PTX). For rapamycin, 71-fold reduced IC_50_ than anative drug with targeted magnetic NPs. EGFR-conjugated immuno NPs exhibited ~13-fold higher uptake and antiproliferative activity than unconjugated NPs in MCF-7 BC cells. HER2-targeted PLGA/montmorillonite-trastuzumab NPs exhibited 12.74 and 13.11-fold larger therapeutic efficacy than untargeted NPs.
Folate receptor	Binds to water-soluble, low-molecular-weight FA (vitamin B9), essential in normal mitotic cell division;, binds to the folate ligand and then internalizes the complexes via receptor-mediated endocytosis.	Studied for high anticancer efficacy of PTX (in 4T1 BC cells), cisplatin (DDP) and docetaxel (in MDA-MB-231 BC cells), CURC (in MCF-7 cells), enhanced cellular uptake for deoxycholic acid-O-carboxymethylated CTS and vincristine sulfate-loaded PLGA-PEG NPs
Estrogen receptors	Belong to thenuclear hormone receptor superfamily, differentially overexpressed in 60–80% of BC cells, internalized in the cancer cells upon binding to ERs	Enhanced cellular uptake of PEG-conjugated PTX-epirubicin co-loaded liposomal NPs, TAM surface-grafted liposomes loaded with DOX (in MCF-7 cells), DOX and DOX + TAM-loaded liposomes (MCF-7 cells)
CD44/Hyaluronan receptor	A natural component of ECM, hyaluronan is critically involved in cell proliferation, migration and invasion. CD44 regulates lymphocyte adhesion to endothelial cells during lymphocyte migration (a process equivalent to metastasis in solid tumors). CD44 prevails as an HA receptorandprevails sparsely on the surface of epithelial, hematopoietic and neuronal cells. Recent studies have screened CD44 as a BC stem cell marker.	HA-functionalized CTS lipoic acid NPs loaded with 17α-methyltestosterone exhibited enhanced internalization in CD44 overexpressing BT-20 BC cells than CD44-negative MCF-7 cells. Enhanced uptake and cytotoxicity of PLGA NPs in MDA-MB-231 cells. Higher docetaxel uptake for self-assembled PLGA-HA block copolymers (in MDA-MB-231 cells) than untargeted NPs. HA matrix NPs with intrinsic CD-44 tropism and loaded rapamycin exhibited a 3.2-fold enhanced uptake in CD44-positive MDA-MB-468 cells. HA-lysine-lipoic acid NPs loaded with DOX exhibited a 20-fold enhanced uptake in MCF-7/ADR cells.
Luteinizing hormone-releasing hormone receptor (LHRH)	A hormonal decapeptide produced by the hypothalamus, LHRH is also known as a gonadotropin-releasing hormone. Plays a key role in regulating the pituitary–gonadal axis and reproduction. Overexpressed in 50% of BC cells.	LHRH-conjugated PTX showed enhanced specificity for targeting MDA-MB-231 cells. LHRH-CTS-conjugated NPs enhanced cellular uptake with two-fold reduced IC_50_in LHR overexpressing MCF-7 cells than non-targeted NPs. DDP-loaded LHRH-modified dextran NPs exhibited a higher uptake with reduced nephrotoxicity of DDP than non-targeted NPs (in 4T1 BC cells).

CTS: chitosan, DDP: cisplatin, CURC: curcumin, PTX: paclitaxel, BC: breast cancer, NPs: nanoparticles, TAM: tamoxifen.

**Table 2 pharmaceutics-15-00227-t002:** Summary of few research attempts using mesoporous silica nanoparticles as delivery vehicles for trafficking chemotherapeutic drugs to bresat **cancer cells.** The in vivo conduct of these investigations infers a readiness for their furtherance to clinical or terminally ill patients [160,206,207,208,209,210].

Chosen Drug(s), Examined Cell Line, Animal Model	Functionalizing Moiety on MSNPs, Prominent Physicochemicaldistinctions and Other Observations	Outcomes of Therapeutic Efficacy
**Studies in singular mode**		
Anderson-type manganese polyoxomolybdate (POMo), MDA-MB-231 cells	PDA coating and glucosamine anchoring, Hydrodynamic size (HS): 195 nm, *ζ*-potential: −18.9 mV, 45% loading	pH-sensitive release profile enhanced anticancer activity compared to the free P_O_Mo with the highest cellular uptake and apoptotic extents
DOX, 4T1 murine mammary carcinoma cells, and BALB/c mice implanted with 4T1 tumors, (1–20) μg·mL^−1^ for 6, 12 and 24 h	Disulfide bonds (SS) as redox-sensitive linkers and HA as capping and targeting agents, (242–250) nm hydrodynamic diameter, <0.09 as PDI over 7 days, glutathione (10 mM, redox sensitivity) and hyaluronidase (150 U·mL^−1^, enzyme sensitivity) presence catalyzed a 58.39% DOX release at <7 pH	Higher uptake and lower IC_50_ of NC delivered DOX in 4T1 (0.39 μg·mL^−1^) than 293T cells (2.83 μg·mL^−1^). SS and HA coating revealed 1.45 (6 h), 1.04 (12 h) and 2.65 (24 h)-fold higher fluorescence than uncoated and non-SS hollow MSNPs
**Combinatorial delivery mode**		
DOX and chlorine Ce6 (Ce6) co-loaded into the pores of -SS-surface conjugated MSNPs. The final coating withcarboxymethylCTS gave DOX/Ce6@MSC NCs, 20 mg DOX with 20 mg Ce6 (1.1:1) administered to MCF-7 human BC cells, monitored for 2, 6 and 12 h for uptake studies	Average PS of MSNPs was 158.07 nm, while for DOX/Ce6@MSC NCs, the *ζ*-potential changes from −37 mV (MSNPs) to 19.1 mv (-NH_2-_functionalized state) and again −32.3 mV for DOX + Ce6-loaded state, implyinga modification, high SA (799.12 m^2^·g^−1^) and PV (0.27 cm^3^·g^−1^) for MSNPs were reduced to 91.72 m^2^·g^−1^and 0.09 cm^3^·g^−1^ on DOX + Ce6 loading. H&E staining showed no major tissue damage in healthy mice until 30 days, non-toxicity of NC	Reduced viability for DOX + Ce6-loaded MSNPs with laser irradiation than DOX alone, DOX + Ce6 delivered via SS-COOH-functionalized and neat NCs. 58.32% survival for DOX + Ce6 delivery but for laser combination, it was only 39.59%. ROS generation was showedby the bright green fluorescence andflow cytometry studies. Successful replication in subcutaneous mice xenografts screened via the highest TV reduction
ATO andPTX, MCF-7 cells, mice bearing MCF-7-derived tumors, 1 mg·kg^−1^ ATO and 0.5 mg·kg^−1^ PTX were administered to MCF-7 tumor-bearing mice, via tail vein every 2 days. Effects were monitored viaTV and body weight measurement on alternate days untilthe 24-day treatment	Coating with PAA and pH-sensitive lipids, tumor-targeting peptide F56, spherical morphology and 117.7 ± 1.51 nm PS (via TEM), which increased to 124.3 ± 1.53 nm for –NH_2_ functionalization, which also increased ζ-potential from −30.3 ± 1.15 to 37.3 ± 1.16 mV.3 nm pore size, 1.088 cm^3^·g^−1^, PV and 1203.453 m^2^·g^−1^ SA, reduced on –NH_2_ functionalization and PAA grafting, 76% ATO and 97% PTX release at <5 pH	1:1, 1:2 and 2:1 PTX + ATO exhibited synergism with <1 CI, 29.5% and 29.9% cell cycle arrest at G2/M phase, 21.45 and 61.48% post 24 h apoptosis for 1:2 and 2:1 proportions, significantly higher than PTX (7.41%) andATO (7.41%), greater Bcl-2inhibition, caspase-7, caspase-9, cyclin B-1 and cyclin D-1 involvement, insignificant effects on singular treatment
Niclosamide (NIC) (Wntsignalinginhibitor) andDOX. Efficacy screened in MDA-MB-231, SKBR3 and MCF-7 BC cells. NIC: (0–128) μM, DOX: (0–64) μM, monitored for 24 h	The combination resulted in synergistically enhanced cell death, both in sequential andconcurrent treatment regimes, feasible for all clinical BC subtypes. Suppressed Wnt/β-catenin signaling with Go/G1 cell cycle arrest (by NIC) and aggravated ROS (both NIC and DOX) alongside the native DOX toxicity, contributed to synergistic response for both treatment regimes	For MDA-MB-231 cells:Of the 56 concurrent combinatorial stoichiometries, 47 caused >50% cell death. All DOX extents with intermediate to high NIC caused >70% cytotoxicity. Of the 47 combinations, 33 were highly synergistic (CI < 0.5). For SKBR3 and MCF-7 cells: All NIC dosages with intermediate to high (SKBR3), and high (MCF-7) numbers of cells were effective. 28 combinations screened in SKBR3 cells, with 15 synergistic outcomes. 6 of 49 combinations in MCF-7 cells revealed a synergism with 19–36% cell death.
siRNA (siPlk1) and miRNA (miR-200c) simultaneous delivery via encapsulated indocyanine green (ICG) to enable stealth trafficking alongside surface conjugation of iRGD peptide MDA-MB-231 TNBC cells siRNA andmiRNA: 100–200 nM, ICG: 0.96–1.92 μg·mL^−1^ monitored until 48 h	91% EE for ICG, variations in *ζ*-potential (34.3 mV for –NH_2_-functionalized MSNPs, which decreased to 21.6 mV on loading of RNA and ICG cargos that further decreased (due to phospholipids and PEGylated lipids) to −19 mV implied successful siRNA + miRNA loading and lipidic surface stabilization	The photodynamic effect of ICG generated ROS and disrupted the endolysosomal membranes, succeeded by the liberation of entrapped therapeutic RNAs. iRGD conjugation augmented the infiltration of encapsulated siRNA + miRNA cargos. MSNPs loaded with Plk1 (siRNA), 200c(miRNA) and ICG combined with light irradiation suppressed the tumor growth to maximum, reducing the lung metastasis by 40% compared to 100% of all other combinations
Combining PTX-loaded mixed micelles (as gating agents) with DOX-loaded MSNPs viapH-sensitive Schiff base	Argine-glycine-aspartic acid (RGD) peptide was used as a targeting agent, with 81% PTX and 65% DOX release on 72 h monitoring	Though apoptotic induction was screened exact quantitative aspects could not be ascertained due to recent online availability.

The studies appear in order of their numbered references with captions. Abbreviations: TV: tumor volume, PAA: polyacrylic acid, ATO: arsenic trioxide, PTX: paclitaxel. SA: surface area, PV: pore volume, CI: combination index, DOX: doxorubicin, MSNPs: mesoporous silica nanoparticles, HA: hyaluronic acid, SS: disulfide linkage, ROS: reactive oxygen species, Ce6: chlorin Ce6 PTX: paclitaxel, PDA: polydopamine, HS: hydrodynamic size, TNBC: triple-negative breast cancer, siRNA: small interfering RNA, miRNA: micro RNA, EE: encapsulation efficacy.

## Data Availability

No data listed in the manuscript were obtained from experimental studies.

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
