# Peer review of "Recent Advances in Mesoporous Silica Nanoparticle-Mediated Drug Delivery for Breast Cancer Treatment"

_pharmaceutics, 2023, doi:10.3390/pharmaceutics15010227_

Round 1

Reviewer 1 Report

This review paper describes the recent advancements in the application of mesoporous silica nanoparticles for breast cancer treatment.

·        The review paper is well written.

·        The study has scientific soundness.

·        This review paper can significantly contribute to the research field.

However, there are minor concerns as

1.       The authors should replace all the old, dated references with the latest research to make their paper more up-to-date.

2.       The authors should clearly mention in the introduction part why they selected silica nanoparticles to write a review, as there are many other new technologies and other new materials being used for breast cancer treatment.

3.       The authors should make another section to describe the challenges in application and future trends.

Author Response

Comment 1: The authors should replace all the old, dated references with the latest research to make their paper more up-to-date.

Response: Now the references earlier to 2005 have been suitably replaced with the studies conducted after 2005. Reference number 36, 47, 75, 120, 121, 123 have been changed and replaced with recent studies or reports. Some references were explicit for the quoted information and particular observations (being recalled by investigators name for the particular findings), that’s why we did not find their suitable replacements. For instance, references 37 to 40 could not be changed as these highlighted initial conceptualization of mesoporous silica nanoparticles.  

Comment 2: The authors should clearly mention in the introduction part why they selected silica nanoparticles to write a review, as there are many other new technologies and other new materials being used for breast cancer treatment.

Response: Now the justification for choosing mesoporous silica nanoparticles is included in the Introduction. The concurrent emergence of Au and Ag nanoparticles and our multiple contributions to Au and Ag nanoparticles encouraged us to choose some other carriers, having achieved similar distinctions.  

Comment 3: The authors should make another section to describe the challenges in application and future trends.

Response: Now, a separate section discussing the future challenges as highlighted in latest studies (related to aqueous degradation of silica and biocompatibility issues for cardiovascular diseases) is included in the manuscript.

Reviewer 2 Report

Comments and suggestions

1.    As the emulsions/ liposomes are nano-carriers available as droplets/vesicles. So, in Figure 2a, rather than using the term “nanoparticles”, it would be better to use the term nanocarriers or nanosystem.

2.    All the sections of this article are too much lengthy. Too much lengthy sections are not good for the readers. They may lose the interest to read the article with flow. These must be reduced with keeping in mind the length of the manuscript and total number of words.

3.    As this article focus on the utilization of MSN as carrier for breast cancers. Therefore, the sections 2 to 4 are not much related, so these must be described in short as possible.

4.    There are repetitions of information, the authors must read the article carefully during its revision.

5.     Section 5 and section 6 are the main part of this article as the title pf the article. So, focus on these two sections in detail.

6.    The vertical lines in the tables (in most of the journals) are not recommended, so remove the vertical lines.

7.    6.2.1. Select studies from 2021, what does it mean?

8.    Mind the margin of the pages, as we can see that the caption of Table 2 “Summary of certain studies employing mesoporous silica nanoparticles as delivery vehicles for delivering chemotherapeutic drugs to breast cancer cells (or possible animal models). The in vivo conduct of these investigations infers a readiness for their furtherance to clinical or terminally ill patients [201, 155, 202-206]” is going out of page.

9.    I suggest the authors to discuss the in vitro drug release and release kinetics from amino-functionalized MSN. Discuss their findings with some published literature.

For example:

a). Pharmaceutics 2020, 12, 1035; doi:10.3390/pharmaceutics12111035)

b) Pharmaceutics 2022, 14(6), 1184; https://doi.org/10.3390/pharmaceutics14061184)

10. I would suggest the authors to make a bookmark of the sections and sub-sections and include it before introduction while submitting the revision of the article.

Author Response

Comment 1: As the emulsions/ liposomes are nano-carriers available as droplets/vesicles. So, in Figure 2a, rather than using the term “nanoparticles”, it would be better to use the term nanocarriers or nanosystem.

Response: Now, the suggested modification has been made in Fig.2(a).

Comment 2: All the sections of this article are too much lengthy. Too much lengthy sections are not good for the readers. They may lose the interest to read the article with flow. These must be reduced with keeping in mind the length of the manuscript and total number of words.

Response: Best efforts have been made to reduce the length and make the illustration easy to understand. For instance, the sections 2 to 4 have been shortened as per the suggestions of Comment 3. However, the essential aspects like special attributes of MSNPs as drug carriers and drug release mechanisms could not be skipped. We have described the last five year studies (since 2017), which has made the explanation a little large. Despite that, we have tried to be brief wherever possible.

Comment 3: As this article focus on the utilization of MSN as carrier for breast cancers. Therefore, the sections 2 to 4 are not much related, so these must be described in short as possible.

Response: Now, sections 2 to 4 of the manuscript have been revised and reduced in length as per the suggestion.

Comment 4: There are repetitions of information the authors must read the article carefully during its revision.

Response: We apologize for these lapses. In the revised version of the manuscript, best attempts have been made to improve the content of the manuscript and wherever possible, the text has been cut short and the abbreviations have been checked for no repetitive mention with their full forms.

Comment 5: Section 5 and section 6 are the main part of this article as the title of the article. So, focus on these two sections in detail.

Response: Keeping in mind the suggestion of Comment 9, the best attempts have been made to improve the sections 5 and 6.

Comment 6: The vertical lines in the tables (in most of the journals) are not recommended, so remove the vertical lines.

Response: Now, the vertical lines have been removed from the tables.

Comment 7: Select studies from 2021, what does it mean?

Response: This heading implies that the corresponding section discusses the 2021 research attempts, involving the use of MSNPs as drug carriers for breast cancer treatment. Now the heading words have been changed to: “Research attempts from 2021”

Comment 8: Mind the margin of the pages, as we can see that the caption of Table 2 “Summary of certain studies employing mesoporous silica nanoparticles as delivery vehicles for delivering chemotherapeutic drugs to breast cancer cells (or possible animal models). The in vivo conduct of these investigations infers a readiness for their furtherance to clinical or terminally ill patients [201, 155, 202-206]” is going out of page.

Response: Now, the margin restrictions have been followed uniformly and the alignment of the Table 2 heading has been properly adjusted.

Comment 9: I suggest the authors to discuss the in vitro drug release and release kinetics from amino-functionalized MSN. Discuss their findings with some published literature.           For example:

 a). Pharmaceutics 2020, 12, 1035; doi:10.3390/pharmaceutics12111035)

  1. b) Pharmaceutics 2022, 14(6), 1184; https://doi.org/10.3390/pharmaceutics14061184)

Response: The discussion of these studies is now included in section 3, sub-section (vii) Drug release from MSNPs, corresponding to reference number 127 and 130

Comment 10: I would suggest the authors to make a bookmark of the sections and sub-sections and include it before introduction while submitting the revision of the article.

Response: Now, the Table of Contents (TOC), summarizing the paper’s contents are included after the Introduction.